# Global multi-ancestry genome-wide analyses identify genes and biological pathways associated with thyroid cancer and benign thyroid diseases

Thyroid diseases are common and highly heritable. We performed a meta-analysis of genome-wide association studies from 19 biobanks for five thyroid diseases: thyroid cancer (ThC), benign nodular goiter, Graves' disease, lymphocytic thyroiditis and primary hypothyroidism. We analyzed genetic association data from ~2.9 million genomes and identified 313 known and 570 new independent loci linked to thyroid diseases. We discovered genetic correlations between ThC, benign nodular goiter and autoimmune thyroid diseases ($rg$ = 0.16–0.97). Telomere maintenance genes contributed to benign and malignant thyroid nodular disease risk, whereas cell cycle, DNA repair and damage response genes were associated with ThC. We propose a paradigm that explains genetic predisposition to benign and malignant thyroid nodules. We found polygenic risk score associations with ThC risk of structural disease recurrence, tumor size, multifocality, lymph node metastases and extranodal extension. Polygenic risk scores identified individuals with aggressive ThC in a biobank, creating an opportunity for genetically informed population screening.

Thyroid diseases are highly prevalent. According to the American Thyroid Association (ATA), over 12% of the US population develops a thyroid condition during their lifetime (www.thyroid.org/media-main/press-room/). Thyroid cancer (ThC) is the most common endocrine malignancy, with 44,020 new cases and 2,170 deaths in the United States in 2024 (ref. 1). Thyroid function diseases, hypothyroidism and hyperthyroidism, negatively affect most organ systems and are associated with disproportionate cardiovascular mortality[2]. It is not well understood why some individuals develop thyroid disease, although genetic[3,4] and environmental factors, such as radiation exposure[5], have a role.

Genetic effects are estimated to contribute up to 53% to ThC susceptibility in family studies[3,4], making ThC one of the most heritable common cancers[3,6]. For autoimmune thyroid diseases, genetic factors account for approximately 75% of the total phenotypic variance[7].

Ruling out thyroid malignancy is a common clinical task because of the high prevalence of thyroid nodules. Thyroid ultrasound reveals nodules in up to 65% of the general population[8,9]. Clinical providers assess thyroid nodule sonographic characteristics[10] to decide if a fine-needle aspiration (FNA) biopsy is necessary. Over 600,000 FNAs are performed annually in the United States to rule out cancer[11], and most (~92%) produce benign, inadequate or indeterminate results[12,13]. Genetic ThC risk assessment with polygenic risk score (PRS) provides an opportunity to improve the diagnostic yield of FNA and reduce unnecessary procedures, molecular tests and diagnostic surgeries[14].

Some ThCs are aggressive, with extensive local invasive growth and distant metastases, leading to ~45,600 deaths annually worldwide[15]. Diagnosing aggressive ThC early, when it can be cured with neck surgery and radioactive iodine[16], can dramatically decrease mortality from the disease. A test to identify individuals at risk of aggressive ThC has not yet been developed. This motivated us to study PRS associations with the high-risk features of ThC.

Discovering genetic variants predisposing to ThC and benign thyroid conditions helps in understanding the biological processes leading to disease. Several genome-wide association studies (GWAS)

✉ e-mail: chris.gignoux@cuanschutz.edu; nikita.pozdeyev@cuanschutz.edu

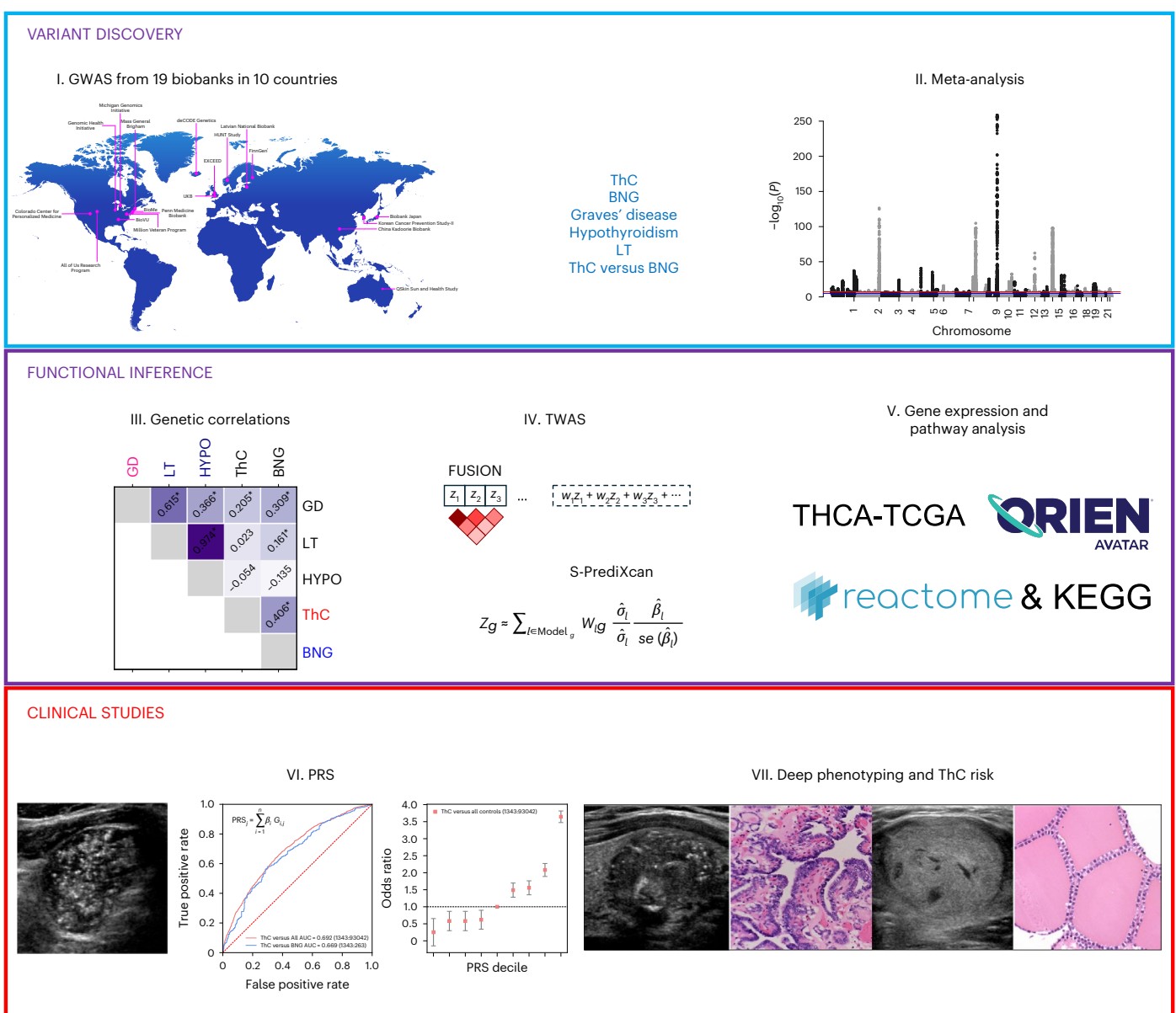

**Fig. 1 | Study design.** I. The VTB Consortium was established within the framework of the GBMI. The participating biobanks performed GWAS for five thyroid diseases. II. An inverse-variance-weighted meta-analysis was conducted after quality control procedures. Previously known and new independent genetic associations were identified. III. Functional inference studies included genetic correlation analysis with cov-LDSC. Asterisks denote Benjamini–Hochberg false discovery rate (FDR) < 0.05. IV. TWAS (FUSION and S-PrediXcan). V. Pathway (KEGG and Reactome) and gene expression analyses (TCGA and ORIEN AVATAR). VI. PRS were developed for ThC, benign thyroid diseases and to distinguish malignant and benign thyroid nodules. VII. PRS were tested for association with thyroid diseases and aggressive ThC features extracted from clinical charts and surgical histopathology reports.

have been conducted on ThC[17–22]. Most recently, the Global Biobank Meta-analysis (GBMI) Consortium combined data from 6,699 individuals with ThC and ~2.2 million controls[23]. GWAS for benign thyroid diseases and related traits, such as thyroid-stimulating hormone (TSH) levels, have been performed in large biobanks, including the UK Biobank (UKB)[24], FinnGen[25], Million Veteran Program[26] and others[27]. However, a systematic analysis of underlying genes, pathways and clinical relevance is missing.

Platforms such as the GBMI (www.globalbiobankmeta.org/ (ref. 23)) enable global collaborations among dozens of participating biobanks, resulting in unmatched GWAS discovery power and data diversity, particularly relevant to cross-phenotype investigations. In this study, we report results from a GBMI project dedicated to thyroid diseases.

## Results

The study had three phases (Fig. 1): (1) variant discovery: GWAS, quality control procedures and meta-analysis; (2) functional inference: genetic correlations, transcriptome-wide association studies (TWAS), pathway and gene expression analyses; and (3) clinical studies: PRS development, testing on the clinical use case of distinguishing benign from malignant thyroid nodules, testing for associations with cancer aggressiveness and testing the utility of PRS for aggressive ThC screening.

### Virtual Thyroid Biopsy Consortium

We founded the Virtual Thyroid Biopsy (VTB) Consortium (Extended Data Fig. 1) under the GBMI (www.globalbiobankmeta.org/)[23] to study the genetic architecture of thyroid diseases at a global multi-ancestry

scale. The Consortium aggregates data from 19 biobanks in ten countries and four continents (Supplementary Table 1). Biobanks performed multi-ancestry or ancestry-stratified GWAS for five thyroid diseases: ThC, benign nodular goiter (BNG), Graves' disease (GD), lymphocytic thyroiditis (LT) and primary hypothyroidism. In addition, a GWAS of ThC versus BNG was performed, focusing on the common clinical task of determining malignancy in thyroid nodules. Phenotype and GWAS definitions are listed in Supplementary Tables 2 and 3.

## Meta-analysis of GWAS

The meta-analysis aggregated data from 198 GWAS summary data files (Supplementary Table 4). Individual GWAS runs were well controlled for confounding (covariate-adjusted linkage disequilibrium score regression (cov-LDSC)[28] $y$ axis intercept $1.00 \pm 0.05$ (mean $\pm$ s.d.)). Healthcare system-based biobanks had a higher disease prevalence than population-based biobanks (Extended Data Fig. 2), as reported previously[23]. Bio*Me*, the All of Us Research Program (AoU) and the Million Veteran Program biobanks had the most diverse participant pools measured using Summix2 (ref. 29).

The meta-analysis included 21,816 cases of ThC, 68,987 cases of BNG, 18,719 cases of GD, 18,331 cases of LT, 257,365 cases of primary hypothyroidism and ~2.9 million controls (Supplementary Table 3). Population structure was determined with Summix2 (ref. 29) via mixture modeling of study-based allele frequencies compared to the gnomAD reference panel[30]. Seventeen percent of genotypes were from individuals of African (AFR-like), 4.4% from Admixed American (AMR-like), 8.1% from East Asian (EAS-like) and 70.5% from European (EUR-like) ancestries.

We found 883 independent loci significantly ($P \leq 5 \times 10^{-8}$) associated with thyroid diseases, including mixed-ancestry and ancestry-stratified genetic associations (Supplementary Tables 5, and 6.1–6.6 and Supplementary Fig. 1). Of these, 313 variants were reported to the NHGRI-EBI Catalog[31] for thyroid traits (as of April 2024); 570 loci were new. Most lead variants were intronic ($n = 407$), followed by intergenic variants ($n = 302$). Among 46 significant exonic variants, 43 were nonsynonymous, potentially altering protein function.

The ancestry-stratified GWAS replicated many associations from the mixed-ancestry meta-analysis and discovered many additional associations (Supplementary Tables 6.1–6.6; variant IDs are indicated by asterisks, $n = 148$). For example, a rare (minor allele frequency (MAF) = 0.0007) nonsynonymous exonic variant in the shelterin complex gene *TERF1* (8:73046129:G:A, $\beta = 1.32$, $P = 1.08 \times 10^{-9}$) was significantly associated with ThC only in the EUR-like meta-analysis (mixed-ancestry GWAS $\beta = 1.16$, $P = 5.5 \times 10^{-4}$). Another plausible EUR-like meta-analysis association is *DIO1* in hypothyroidism (1:53909897:C:A, $\beta = -0.024$, $P = 5.06 \times 10^{-11}$). *DIO1* encodes an enzyme that converts pro-hormone thyroxine to the active thyroid hormone tri-iodothyronine[32].

## Single-nucleotide polymorphism heritability and genetic correlation

The cov-LDSC-estimated $h^2_{SNP}$ ranged from 0.07 (s.e. = 0.01) for BNG in the mixed-ancestry meta-analysis to 0.11 (0.01) for the mixed-ancestry hypothyroidism meta-analysis (Supplementary Table 7).

There was a strong genetic correlation between LT and hypothyroidism (mixed-ancestry, $rg = 0.97$ (0.04), $P = 2.05 \times 10^{-106}$, Fig. 2 and Supplementary Table 8). We found significant (Benjamini–Hochberg false discovery rate (FDR) < 0.05) genetic correlations between LT and GD ($rg = 0.62$ (0.09)), LT and BNG ($rg = 0.16$ (0.07)), ThC and BNG ($rg = 0.41$ (0.16)), GD and hypothyroidism ($rg = 0.37$ (0.07)), GD and BNG ($rg = 0.31$ (0.07)), and GD and ThC ($rg = 0.20$ (0.05)). Genetic correlation analysis in the EUR-like meta-analysis yielded similar results (Extended Data Fig. 3 and Supplementary Table 8).

## TWAS

We performed a *cis*-acting expression quantitative trait locus (*cis*-eQTL) TWAS using two methods, FUSION[33] and Summary-based PrediXcan (S-PrediXcan)[34,35], and GTEx v.8 thyroid tissue expression models[36], to identify potential causal variants affecting gene expression and assign intergenic and noncoding RNA variants to protein-coding genes.

The FUSION TWAS, as applied to the mixed-ancestry and EUR-like ThC GWAS meta-analysis, identified the expression of 55 unique protein-coding genes (Supplementary Tables 9.1 and 9.2). FUSION also identified 47 and 45 significant (after Bonferroni adjustment) lead *cis*-eQTL variants from the mixed-ancestry and EUR-like GWAS, respectively. The TWAS attributed many significant intergenic and noncoding variants to protein-coding genes based on reported eQTL status. For example, noncoding RNA intronic variant 1:218515813:T:C (mixed-ancestry ThC GWAS meta-analysis $P = 4.07 \times 10^{-39}$) was attributed to the expression of *TGFB2* in the TWAS ($P = 3.59 \times 10^{-61}$). Most significant genes found by the FUSION TWAS were also replicated by S-PrediXcan, indicating the analytical rigor of our analyses (Supplementary Tables 6.1–6.6).

Consistent with a genetic overlap between thyroid diseases (Fig. 2, Extended Data Fig. 3 and Supplementary Table 8), we found that many genes were discovered in more than one thyroid phenotype TWAS (Supplementary Table 9.3). For example, *cis*-eQTLs and expression of *TGFB2* were associated with all thyroid diseases in our analysis and the TSH trait[37]. Plausibly, most overlap in the TWAS analyses was between autoimmune thyroid diseases and TSH[37], a hormone that is clinically measured to diagnose hypothyroidism and GD (Supplementary Table 9.4).

The TWAS found additional significant genes where the GWAS meta-analysis failed to identify genome-wide significant associations, for example, *VEGFC* ($P = 1.30 \times 10^{-6}$) and *NBR1* ($P = 1.02 \times 10^{-6}$), further expanding our knowledge of genes associated with ThC risk.

## Gene expression analysis

We evaluated the mRNA expression of genes discovered in the ThC GWAS meta-analysis and the TWAS in normal and malignant thyroid tissues (Extended Data Fig. 4 and Supplementary Table 10). Of the 20 evaluated tissues[38], normal thyroid tissue was among the top three highest-expressing tissues for 20 genes. Two genes, *TG* and *NKX2-1*, are expressed only in the thyroid.

The expression of six genes (*ETS1*, *HMGA2*, *NFIA*, *PCNX2*, *PIBF1* and *VAV3*) significantly correlated with younger age at ThC diagnosis in The Cancer Genome Atlas (TCGA) study for papillary ThC (THCA-TCGA)[39] or the Oncology Research Information Exchange Network (ORIEN) AVATAR study (www.oriencancer.org/) (Bonferroni-corrected $P \leq 1.08 \times 10^{-4}$). *TERT* expression correlated with older age at diagnosis ($P = 1.6 \times 10^{-8}$), matching a similar association with somatic *TERT* promoter mutations[40]. The expression of 23 genes was positively correlated with at least one clinical or molecular ThC risk feature: younger age at diagnosis, higher stage, presence of extrathyroidal extension, lower BRAF/RAS score (indicating a BRAF-like expression profile[39]), higher ERK score (measuring RAS/MAPK pathway activity) and lower ThC differentiation (estimated with thyroid differentiation score[39]; Extended Data Fig. 4 and Supplementary Table 10).

## Pleiotropic and disease-specific associations

**ThC and BNG.** We do not know why some patients develop BNGs while others get ThC. To understand the cellular functions and pathways leading to benign or malignant thyroid nodular disease, we explored pleiotropic, and ThC-specific and BNG-specific, loci (Fig. 2).

We generated locus plots for independent lead variants from the GWAS meta-analysis (Supplementary Fig. 2.1–2.3). We categorized loci and genes as those significantly associated with: (1) ThC but not BNG (may contribute to malignant transformation of

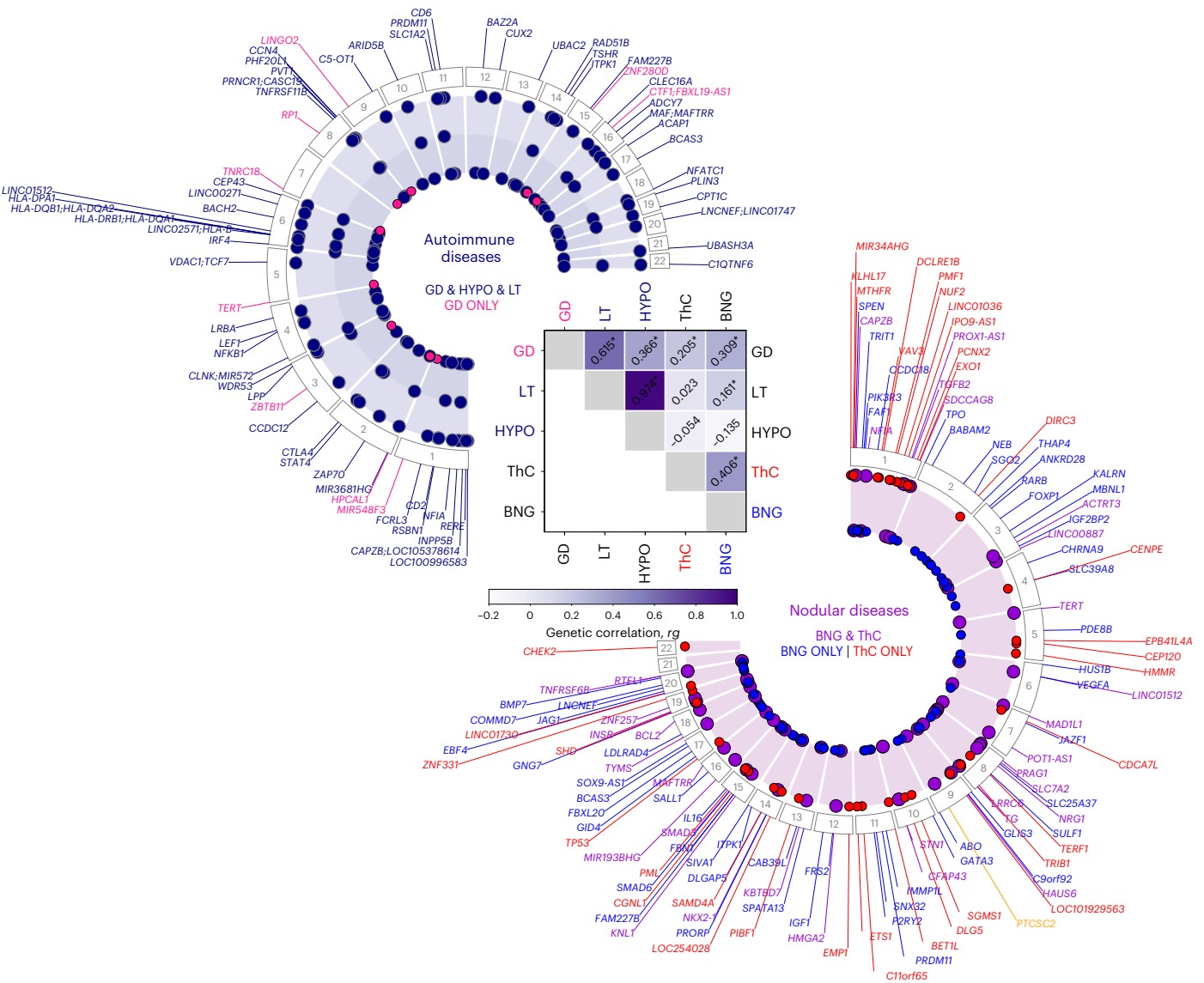

**Fig. 2 | Pleiotropic and phenotype-specific loci associated with thyroid diseases in the meta-analysis of GWAS.** The heatmap illustrates the genetic correlation (*rg*) between thyroid phenotypes, which was estimated using cov-LDSC. The asterisks denote significance at a Benjamini−Hochberg FDR < 0.05. Circular plots highlight loci significantly associated with ThC and BNG (right) and autoimmune thyroid diseases (left). Right, The red and blue dots, along with the gene labels, indicate loci predominantly associated with ThC and BNG, respectively. Left, The red dots indicate loci significantly associated with GD but not with LT or primary hypothyroidism. *PTCSC2* (right, yellow) is the only locus inversely associated with ThC and BNG (Supplementary Tables 6.1–6.6 list all loci).

follicular cells; Supplementary Table 11 and Supplementary Fig. 2.1); (2) BNG but not ThC (may lead to nonneoplastic thyroid nodules and thyroid neoplasms with low malignant potential; Supplementary Fig. 2.2); and (3) both benign and malignant thyroid nodules (Supplementary Fig. 2.3).

Among 36 loci associated with ThC but not BNG, seven are in genes that encode components of cell cycle checkpoints, proteins regulating centrosome and kinetochore function, microtubule attachment and chromosome segregation (*CDCA7L*, *CENPE*, *CEP120*, *CHEK2*, *NUF2*, *PMF1*, *TP53*). The ThC-specific locus *C11orf65* overlaps with the cell cycle checkpoint kinase gene *ATM* (for example, 11:108267276, *ATM* p.Phe858Leu, $P = 4.8 \times 10^{-9}$), which is frequently mutated in advanced ThC[41,42]. Locus *LINC01730* contains variants in the cell cycle regulator gene *CDC25B* (20:3805337:C:T, *CDC25B* 3′UTR, $P = 3.1 \times 10^{-10}$). Loci *HAUS6* (microtubule attachment to the kinetochore and central spindle formation[43]) and *SDCCAG8* (centrosome-associated protein[44]) demonstrate a stronger association with ThC despite the greater statistical power of the BNG GWAS meta-analysis.

Five genes with ThC-specific associations have a role in DNA repair and cellular response to DNA damage (*ATM*, *DCLRE1B*, *PCNX2*, *EXO1*, *TP53*).

BNG-specific loci (*n* = 56) are located in genes participating in insulin-like growth factor 1 (*IGF1* and *IGF2BP2*) and fibroblast growth factor (*FGF7* (*FAM227B locus*) and *FRS2*) signaling pathways. Genes having a role in thyroid gland development and thyroid hormone synthesis were linked to benign nodules (*GLIS3*, *TPO*) but some are also associated with ThC (*NKX2-1* (*LINC00609* locus), *TG*).

Notably, telomere maintenance genes (*ACTRT3*, *LRRC6*, *STN1*, *TERT*) were associated with both ThC and BNG. Genes participating in apoptosis and transforming growth factor-beta signaling are present in all three gene categories (Supplementary Table 11) and contribute to the development of both benign and malignant thyroid nodules. Variants in some of these overlapping genes (for example, TERT, 5:1282299:G:A, $\beta = 0.15$ [0.01], $P$ value = $3.3 \times 10^{-44}$; NRG1, 8:32572853:A:G, $\beta = -0.24$ [0.01], $P$ value = $1.2 \times 10^{-112}$) were also significant in our meta-analysis of thyroid cancer vs. benign

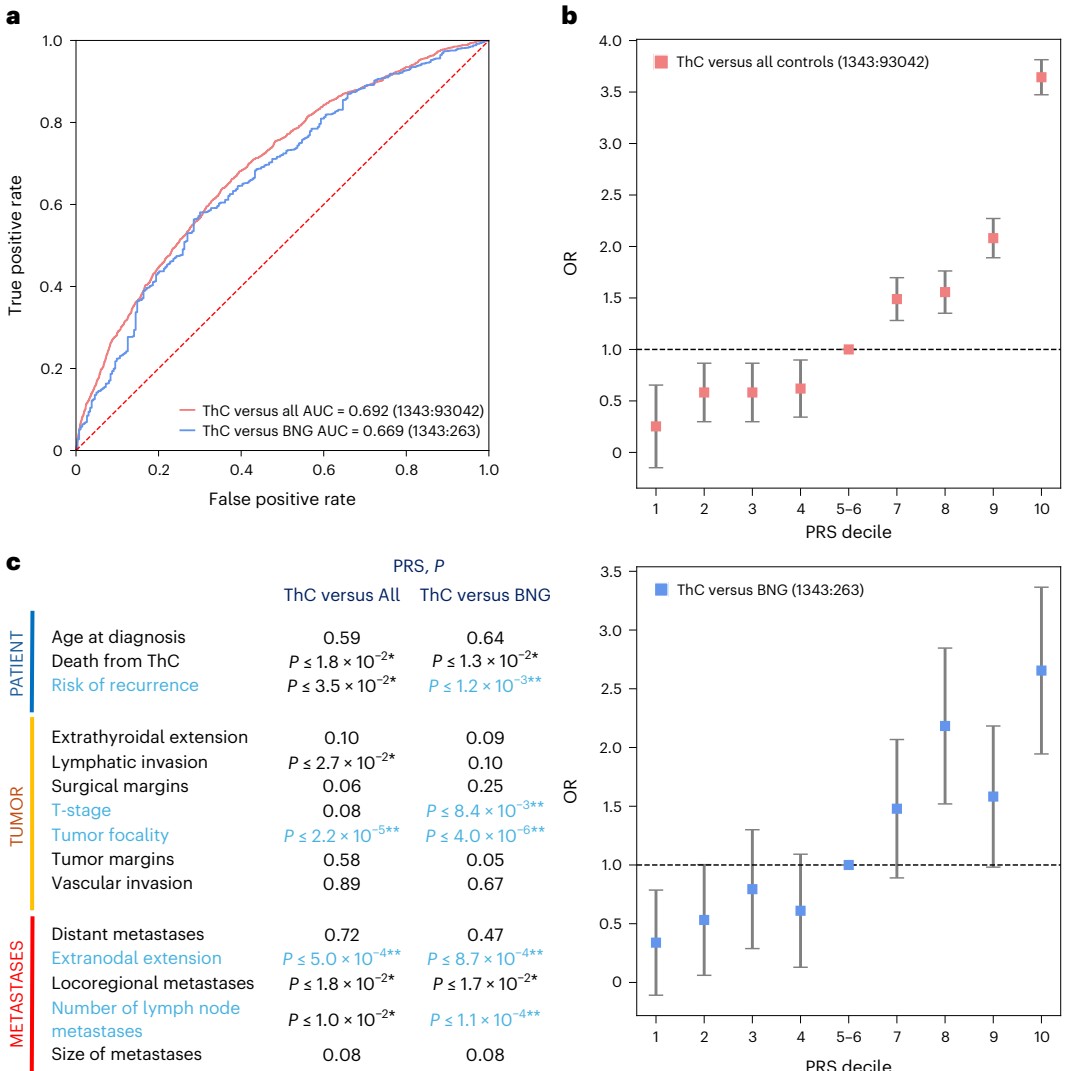

**Fig. 3 | The ThC PRS.** Two ThC PRS were developed: PRS$_{ThC versus All}$ to identify individuals at risk in a population and PRS$_{ThC versus BNG}$ for the clinically relevant task of discriminating malignant and benign thyroid nodules. PRS were tested in the CCPM population, which was not used for PRS development. **a**, AUCs (n = 94,561; 1,343 ThCs). **b**, ThC risk according to PRS decile. The error bars denote the 95% CI calculated as ± s.e. × 1.96 surrounding the odds ratio (OR). **c**, PRS association with features of aggressive ThC. P values were calculated using a two-sided Wald test. Asterisks indicate ThC risk features significantly associated with PRS at a nominal (black; *P ≤ 0.05) or Bonferroni-corrected (blue; **P ≤ 1.7 × 10$^{-3}$) significance threshold. Raw PRS and ThC risk features are listed in Supplementary Table 16.

nodular goiter GWAS (Supplementary Table 6.6), indicating differential contribution to these diseases. Of particular interest is the *PTCSC2* locus because its significant variants have the opposite direction of effect with ThC and BNG (Extended Data Fig. 5, ρ = −0.77, P = 1.2 × 10$^{-24}$).

The Kyoto Encyclopedia of Genes and Genomes (KEGG)[45] and Reactome[46,47] pathway analysis identified cell cycle, senescence and apoptosis as key biological processes contributing to ThC risk (Supplementary Tables 12 and 13). The IGF1 and PI3K/Akt signaling pathways were significantly associated with BNG.

**Autoimmune thyroid diseases.** GD and LT/primary hypothyroidism are related autoimmune endocrine diseases with opposite clinical manifestations, causing hyperthyroidism and hypothyroidism, respectively[48].

Plausibly, for most genes and the KEGG and Reactome pathways associated with GD, LT and hypothyroidism (Supplementary Tables 12 and 13) are related to the immune system. Nine loci, including *CD40*, *LINGO2*, *TNRC18* and *TERT*, were discovered in GD (P < 5 × 10$^{-8}$) but not the hypothyroidism GWAS meta-analysis (Supplementary Fig. 3.1).

Almost all loci significantly associated with GD are also linked to primary hypothyroidism (Supplementary Fig. 3.2). Genetic associations with LT (Supplementary Table 6.4) replicated those with primary hypothyroidism (Fig. 2).

**PRS for ThC diagnosis**
PRS quantifies an individual's risk for developing a specific trait or disease based on genetics. We explored the ability of PRS to identify people at risk for ThC (PRS$_{ThC versus All}$) in the Colorado Center for Personalized Medicine (CCPM) Biobank population (n = 94,651). PRS$_{ThC versus All}$ was calculated from the independent, significant variants identified in the multi-ancestry ThC meta-analysis, excluding CCPM from training to avoid overfitting.

Papillary ThC was the most common thyroid malignancy in the CCPM cohort (n = 1,024), followed by follicular thyroid carcinoma (n = 41), oncocytic thyroid carcinoma (n = 11), anaplastic thyroid carcinoma (n = 7) and poorly differentiated thyroid carcinoma (n = 4). For 253 patients with ThC, the histological subtype was not documented in the clinical records.

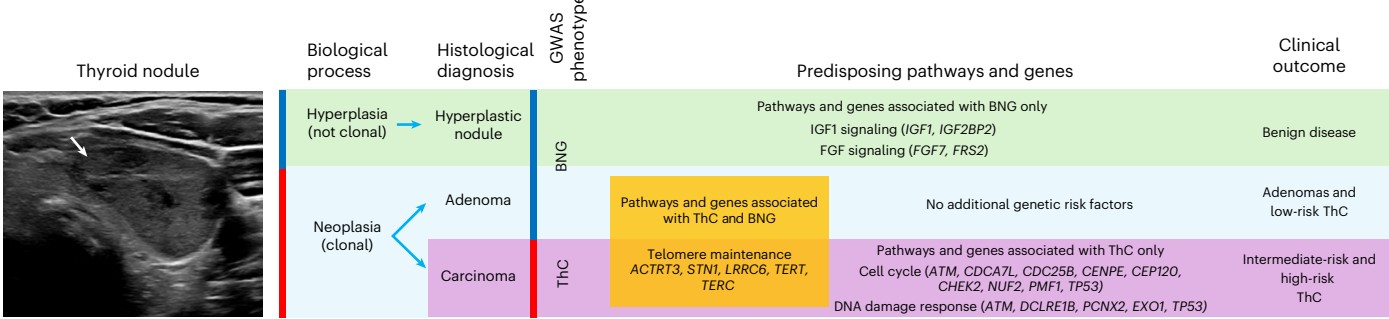

**Fig. 4 | Germline genetic susceptibility to ThC and BNG.** We hypothesize that two biological processes with distinct genetic architecture cause thyroid nodules: (1) hyperplasia, a polyclonal follicular cell proliferation with no malignant potential; and (2) neoplasia, a clonal growth driven by somatic genetic alterations. Neoplastic nodules can be benign or malignant, and the mismatch between biological mechanisms (hyperplasia and neoplasia) and GWAS phenotype definitions (benign and malignant thyroid nodules) has led to apparent genetic pleiotropy. The pathway and genes associated with BNG but not ThC in the GWAS meta-analysis (for example, the insulin-like growth factor 1 (*IGF1*) and fibroblast growth factor (*FGF*) signaling pathways) predispose to benign nodules. Pathways and genes associated with both BNG and ThC (for example, telomere maintenance) predispose to neoplastic thyroid nodules, either benign or malignant. In the absence of other genetic risk factors, patients develop benign adenomas or low-risk ThCs. Alternatively, genetic alterations in cell cycle and DNA damage response genes (associated predominantly with ThC but not BNG in the GWAS meta-analysis) predispose to high-risk ThC.

We assessed the utility of PRS for the clinically relevant task of distinguishing benign from malignant thyroid nodules ($PRS_{ThC\ versus\ BNG}$). $PRS_{ThC\ versus\ BNG}$ was defined as the difference between $PRS_{ThC\ versus\ All}$ and the PRS for BNG ($PRS_{BNG\ versus\ All}$): $PRS_{ThC\ versus\ BNG} = PRS_{ThC\ versus\ All} - PRS_{BNG\ versus\ All}$. We explored the ability of $PRS_{ThC\ versus\ All}$ to identify individuals susceptible to high-risk ThC in a biobank population.

$PRS_{ThC\ versus\ All}$ achieved an area under the curve (AUC) of 0.692 (95% confidence interval (CI) = 0.673 to 0.711; Fig. 3a and Supplementary Table 14). Individuals with $PRS_{ThC\ versus\ All}$ in the top decile had 10.7 times the odds of developing ThC than those in the first decile (Fig. 3b).

Our $PRS_{ThC\ versus\ All}$ significantly outperformed the ThC PRS derived from the previous GWAS meta-analysis from the GBMI phase I project[23] (AUC 0.651 (0.632–0.671), DeLong test $P = 1.01 \times 10^{-10}$) because of the greater discovery power of a large meta-analysis.

To test PRS performance on a clinically relevant use case of discriminating between benign and malignant thyroid nodules (ThC versus BNG), three clinicians (C.C.B., T.L.J. and N.P.) performed clinical chart reviews. We confirmed the diagnosis of non-medullary ThC in 1,343 patients and the diagnosis of BNG in 281. All benign cases were supported by surgical histopathology to avoid contamination because of small ThCs not eligible for biopsy.

$PRS_{ThC\ versus\ All}$ performed worse for the clinical ThC versus BNG task (AUC 0.622 (0.576–0.668)), which was expected because of the genetic associations shared between ThC and BNG. $PRS_{ThC\ versus\ BNG}$, leveraging genetic associations with both ThC and BNG, demonstrated an improved AUC for the ThC versus BNG clinical task (0.670 (0.612–0.728), DeLong test $P = 3.4 \times 10^{-4}$). Thyroid nodules in individuals with $PRS_{ThC\ versus\ BNG}$ in the top decile had 7.8 times the odds of being malignant than in individuals with $PRS_{ThC\ versus\ BNG}$ in the first decile (Fig. 3b).

Benign thyroid disease PRS AUCs ranged from 0.591 (0.580–0.603) for BNG to 0.659 (0.625–0.693) for GD. PRS analyses in the European population showed results similar to those from the mixed-ancestry GWAS meta-analysis (Supplementary Table 14).

Incorporating demographic and genetic ancestry covariates improved predictions for ThC ($PRS_{ThC\ versus\ All}$ AUC 0.725 (0.708–0.742)) and other thyroid diseases (AUC ranging from 0.690 (0.662–0.718) for BNG to 0.729 (0.714,0.745) for hypothyroidism). We expected this improvement because of the higher incidence of thyroid disease in women[49] and the increased risk of developing thyroid nodules and hypothyroidism with age[8,50]. However, no significant improvement in clinical $PRS_{ThC\ versus\ BNG}$ performance was observed (Supplementary Table 14).

We did not find a significant drop in PRS performance measured with AUC in the EAS-like, AMR-like and AFR-like strata (DeLong test, $P > 0.05$) except for hypothyroidism PRS in AFR-like individuals (Supplementary Table 15).

### PRS and ThC aggressiveness

We evaluated associations between ThC PRS and aggressive features of ThC in three domains (patient, tumor and metastatic disease), abstracted from surgical histopathology reports and clinical notes (Fig. 3c and Supplementary Table 16). $PRS_{ThC\ versus\ All}$ was significantly associated with tumor focality and extranodal extension. $PRS_{ThC\ versus\ BNG}$ was significantly associated with the risk of structural disease recurrence (defined according to the ATA guidelines[16]), tumor size (T stage), tumor focality, extranodal extension and the number of neck lymph node metastases (Bonferroni-adjusted $P \leq 1.7 \times 10^{-3}$). At a nominal $P \leq 0.05$ both $PRS_{ThC\ versus\ All}$ and $PRS_{ThC\ versus\ BNG}$ were also associated with death from ThC and locoregional metastases.

To simulate screening for aggressive ThC (high-risk of structural disease recurrence as per ATA[16]), we tested PRS performance when all individuals not diagnosed with high-risk ThC, including those diagnosed with low-risk and intermediate-risk ThC, were considered as controls. $PRS_{ThC\ versus\ All}$ demonstrated a superior AUC of 0.741 (0.682–0.801), sensitivity of 0.803 (0.803–0.803) and specificity of 0.569 (0.565–0.572).

### Discussion

We completed a GWAS meta-analysis for five thyroid diseases, leveraging a global collaboration involving 19 biobanks from ten countries. The Consortium replicated 313 genetic associations deposited in the NHGRI-EBI GWAS Catalog as of April 2024 (v.1.0.2) and discovered 570 new associations (Supplementary Table 5).

Genetic correlation analysis (Fig. 2) identified physiologically plausible and clinically meaningful associations between thyroid diseases. Chronic LT is a leading cause of primary hypothyroidism[51], explaining the near-perfect genetic correlation between these two diseases. The shared genetic basis for LT and GD is expected because both conditions are autoimmune diseases with highly concordant familial risk[52]. The genetic correlation between GD and thyroid nodular disease (both benign nodules and ThC) is mechanistically explained by enhanced TSH receptor signaling, which promotes thyroid epithelial growth and protects thyroid cells from apoptosis[53]. A previous population-based study found an increased risk of thyroid

(hazard ratio = 10–15) and other cancers in patients with GD[54], which is consistent with our findings.

Shared (*rg* = 0.4–0.5) and unique genetic associations with ThC and BNG allowed insights into genes and pathways that lead to malignant and benign thyroid nodules. Our hypothesis explaining why some individuals are susceptible to thyroid nodules while others develop ThC is shown in Fig. 4. We propose that two biological processes with distinct genetic architecture cause thyroid nodules: (1) hyperplasia, a polyclonal follicular cell proliferation with no malignant potential; and (2) neoplasia, a clonal growth driven by somatic genetic alterations. Neoplastic nodules can be benign or malignant, causing the mismatch between biological mechanisms (hyperplasia versus neoplasia) and GWAS phenotype definitions (benign and malignant thyroid nodules), resulting in partial overlap in genetic associations and genetic correlation between ThC and BNG.

We found that genes participating in the cell cycle, DNA repair and cellular response to DNA damage are predominantly associated with ThC but not benign nodules, highlighting the importance of these biological processes for malignant transformation of thyroid follicular cells. These variants and genes can lead to more aggressive multifocal and metastatic ThC. On the other hand, genes in the fibroblast growth factor and insulin-like growth factor 1 signaling pathways were uniquely associated with BNG and may lead to hyperplastic benign thyroid nodules without malignant potential. Variants in genes participating in telomere maintenance increase the risk of ThC and benign neoplastic thyroid nodules (adenomas). Telomere maintenance genes are also associated with syndromic papillary ThC[55,56].

Our finding that autoimmune thyroid disorders share most genetic associations (Fig. 2) indicates that similar fundamental mechanisms lead to GD and LT/primary hypothyroidism despite opposite clinical manifestations.

Of special interest are genes that were only found in the GD meta-analysis despite the greater discovery power of the hypothyroidism GWAS. These genes (*CD40*, *LINGO2*, *TNRC18*, *TERT*) may be involved in immune system processes that define the type of autoantibodies produced: TSH receptor antibodies in GD or thyroid peroxidase/thyroglobulin antibodies in LT and primary hypothyroidism. Consistently, variants in *TSHR* were strongly associated with GD (for example, 14:80990913:A:C, $\beta$ = 0.27 (0.01), $P$ = 2.52 × 10$^{-137}$), while *TPO* and *TG* associations were only seen in the hypothyroidism meta-analysis.

ThC caused 2,170 deaths in the United States in 2024 (ref. 1). PRS derived from the ThC GWAS can identify individuals at ThC risk in the population (Fig. 3a,b and ref. 57). ThC screening is not currently recommended by the US Preventive Services Task Force[58] because of concerns about overtreatment and lack of mortality benefit. However, we found that this PRS is associated with high-risk ThC features (Fig. 3c) and helps discover individuals susceptible to high-risk ThC in a biobank population. The number needed to screen to identify one individual with high-risk ThC in the CCPM cohort was 268. For comparison, the US Preventive Services Task Force-recommended screening for colon cancer with colonoscopy[59] has a number needed to screen of 263 (ref. 60). Thus, genetically informed screening for high-risk ThC is a conceptually viable strategy to identify aggressive ThCs at an early curable stage to reduce morbidity and mortality.

Another clinically meaningful application for the ThC PRS is to aid in the diagnosis of ThC in patients with thyroid nodules[14]. Despite the widespread use of clinical ultrasound-based algorithms[8,9], 72% of FNAs produce benign results and 20% are inadequate or indeterminate[12,13,61,62]. The PRS provides a cancer risk assessment that is complementary and synergistic to ultrasound-based nodule evaluation when combined with computer-vision-based analysis of thyroid ultrasound images[14]. Additional studies of the PRS in combination with clinical thyroid nodule risk stratification algorithms, such as the American College of Radiology Thyroid Imaging, Reporting and Data System[10], are needed.

We found that incorporating variants from both ThC and BNG meta-analysis (PRS$_{\text{ThC versus BNG}}$) improved PRS performance for distinguishing benign and malignant thyroid nodules. Active surveillance of thyroid nodules with low-risk sonographic appearance in patients with reassuring PRS could reduce the need for invasive procedures.

PRS provides a noninvasive risk assessment that is independent of somatic changes used in molecular tests for the management of thyroid nodules with indeterminate cytology[63,64]. Therefore, incorporating PRS is likely to improve the performance of these tests. PRS may guide which biopsy-proven ThCs are likely to be indolent and therefore suitable for active surveillance. It is unknown whether the PRS can improve the assessment of the postoperative risk of recurrence and inform postoperative management.

We recognize that, because of the demographics of participants in the VTB Consortium, we are underpowered in our ability to study individuals of non-EUR-like ancestry. As our Consortium grows, we look forward to conducting more ancestry-specific analyses to ensure that we find the results relevant to all individuals[65] and improve our understanding of rare variation across groups. Fine-mapping analysis will be necessary to discover putative causal variants. The PRS will require calibration and prospective testing in clinical trials before introduction into routine clinical practice.

In summary, we conducted the meta-analysis of the GWAS for five thyroid diseases. We found many previously known and new mechanistically plausible variants, genes and pathways contributing to the risk of ThC, BNG and autoimmune thyroid diseases. We explained why some individuals are prone to developing benign thyroid nodules while others are at risk of multifocal metastatic ThC. We derived and tested PRS for aggressive ThC population screening and for a clinical task of distinguishing benign and malignant thyroid nodules. This study will serve as a foundation for future clinical applications leveraging the germline genetics of thyroid diseases.

## Online content

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

**Samantha L. White** [1], **Maizy S. Brasher**[1], **Jack Pattee**[2], **Wei Zhou** [3,4,5], **Sinéad Chapman** [6], **Yon Ho Jee** [7], **Caitlin C. Bell**[8], **Taylor L. Jamil**[9], **Martin Barrio**[10], **Christopher H. Arehart** [11,12], **Luke M. Evans** [11,12], **Jibril Hirbo** [13,14], **Nancy J. Cox**[13,14], **Peter Straub**[13,14], **Shinichi Namba** [15,16,17], **Emily Bertucci-Richter**[18], **Lindsay Guare** [19], **Ahmed Edris**[20], **Sam Morris** [20], **Ashley J. Mulford**[21], **Haoyu Zhang** [1], **Brian Fennessy** [22], **Martin D. Tobin**[23,24], **Jing Chen** [23], **Alexander T. Williams**[23], **Catherine John**[23,24], **David A. van Heel** [25], **Rohini Mathur**[26], **Sarah Finer** [26], **Marta R. Moksnes** [27,28], **Ben M. Brumpton** [27,29,30], **Bjørn Olav Åsvold** [27,29,31], **Raitis Peculis**[32], **Vita Rovite**[32], **Ilze Konrade**[33], **Ying Wang** [3], **Kristy Crooks**[34], **Sameer Chavan**[34], **Matthew J. Fisher**[34], **Nicholas Rafaels**[34], **Meng Lin**[1,34], **Jonathan A. Shortt** [1,34], **Alan R. Sanders** [21,35], **David C. Whiteman**[36], **Stuart MacGregor**[36], **Sarah E. Medland**[36], **Unnur Thorsteinsdóttir**[37,38], **Kári Stefánsson** [37,38], **Tugce Karaderi**[39], **Kathleen M. Egan**[40], **Therese Bocklage**[41,42], **Hilary C. McCrary**[43,44], **Gregory Riedlinger**[45], **Bodour Salhia**[46,47], **Craig Shriver**[48], **Minh D. Phan**[49,50], **Janice L. Farlow**[51], **Stephen Edge**[52,53], **Varinder Kaur** [54,55], **Michelle L. Churchman** [56], **Robert J. Rounbehler**[56], **Pamela L. Brock** [57], **Matthew D. Ringel**[57,58], **Milton Pividori** [1,34,59], **Rebecca Schweppe**[8,60], **Christopher D. Raeburn**[10], **Robin G. Walters** [20], **Zhengming Chen** [20], **Liming Li** [61,62,63], **Koichi Matsuda** [64,65], **Yukinori Okada** [15,16,17], **Sebastian Zöllner** [18], **Anurag Verma** [19], **Penn Medicine BioBank**\*, **Michael H. Preuss** [22], **Eimear Kenny**[22], **Audrey E. Hendricks** [1], **Lauren Fishbein**[1,8,60,66], **Peter Kraft** [7,67], **Mark J. Daly** [3,4,68], **Benjamin M. Neale** [3,4,68], **Virtual Thyroid Biopsy Consortium**\*, **Colorado Center for Personalized Medicine**[69]\*, **Genes & Health Research Team**[70]\*, **The BioBank Japan Project**[71]\*, **Alicia R. Martin** [3,4,68,72], **Joanne B. Cole** [1,59], **Bryan R. Haugen**[8,60], **Global Biobank Meta-analysis Initiative**\*, **Christopher R. Gignoux** [1,34,59,60] ✉ & **Nikita Pozdeyev** [1,8,34,60] ✉

[1]Department of Biomedical Informatics, University of Colorado Anschutz, Aurora, CO, USA. [2]Center for Innovative Design & Analysis, Colorado School of Public Health, University of Colorado Anschutz, Aurora, CO, USA. [3]Program in Medical and Population Genetics, Broad Institute of MIT and Harvard, Cambridge, MA, USA. [4]Stanley Center for Psychiatric Research, Broad Institute of MIT and Harvard, Cambridge, MA, USA. [5]Center for Genomic Medicine, Massachusetts General Hospital and Harvard Medical School, Boston, MA, USA. [6]The Broad Institute, Cambridge, MA, USA. [7]Department of Epidemiology, Harvard T.H. Chan School of Public Health, Boston, MA, USA. [8]Division of Endocrinology, Diabetes and Metabolism, University of Colorado Anschutz, Aurora, CO, USA. [9]Department of Otolaryngology, Head and Neck Surgery, University of Colorado Anschutz, Aurora, CO, USA. [10]Division of GI, Trauma, and Endocrine Surgery, Department of Surgery, University of Colorado Anschutz, Aurora, CO, USA. [11]Institute for Behavioral Genetics, University of Colorado Boulder, Boulder, CO, USA. [12]Department of Ecology & Evolutionary Biology, University of Colorado Boulder, Boulder, CO, USA. [13]Division of Genetic Medicine, Department of Medicine, Vanderbilt University Medical Center, Nashville, TN, USA. [14]Vanderbilt Genetic Institute, Vanderbilt University Medical Center, Nashville, TN, USA. [15]Department of Genome Informatics, Graduate School of Medicine, The University of Tokyo, Tokyo, Japan. [16]Department of Statistical Genetics, Osaka University Graduate School of Medicine, Suita, Japan. [17]Laboratory for Systems Genetics, RIKEN Center for Integrative Medical Sciences, Yokohama, Japan. [18]Department of Biostatistics, University of Michigan, Ann Arbor, MI, USA. [19]Department of Medicine, Perelman School of Medicine, University of Pennsylvania, Philadelphia, PA, USA. [20]Nuffield Department of Population Health, University of Oxford, Oxford, UK. [21]Genomic Health Initiative, Endeavor Health Research Institute, Evanston, IL, USA. [22]The Charles Bronfman Institute for Personalized Medicine, Icahn School of Medicine at Mount Sinai, New York, NY, USA. [23]Division of Public Health and Epidemiology, University of Leicester, Leicester, UK. [24]University

Hospitals of Leicester NHS Trust, Leicester, UK. [25]Blizard Institute, Queen Mary University of London, London, UK. [26]Wolfson Institute of Population Health, Queen Mary University of London, London, UK. [27]HUNT Center for Molecular and Clinical Epidemiology, Department of Public Health and Nursing, NTNU, Norwegian University of Science and Technology, Trondheim, Norway. [28]Levanger Hospital, Nord-Trøndelag Hospital Trust, Levanger, Norway. [29]HUNT Research Centre, Department of Public Health and Nursing, NTNU, Norwegian University of Science and Technology, Trondheim, Norway. [30]Clinic of Medicine, St. Olavs Hospital, Trondheim University Hospital, Trondheim, Norway. [31]Department of Endocrinology, Clinic of Medicine, St. Olavs Hospital, Trondheim University Hospital, Trondheim, Norway. [32]Latvian Biomedical Research and Study Centre, Riga, Latvia. [33]Department of Internal Medicine, Riga Stradins University, Riga, Latvia. [34]Colorado Center for Personalized Medicine, University of Colorado Anschutz, Aurora, CO, USA. [35]Department of Psychiatry and Behavioral Neuroscience, University of Chicago, Chicago, IL, USA. [36]QIMR Berghofer Medical Research Institute, Herston, Queensland, Australia. [37]deCODE genetics/Amgen, Inc., Reykjavik, Iceland. [38]Faculty of Medicine, University of Iceland, Reykjavik, Iceland. [39]Department of Public Health, Faculty of Health and Medical Sciences, Center for Health Data Science, Section for Health Data Science and Artificial Intelligence, University of Copenhagen, Copenhagen, Denmark. [40]Department of Cancer Epidemiology, H. Lee Moffitt Cancer Center & Research Institute, Tampa, FL, USA. [41]University of Kentucky Markey Cancer Center, Lexington, KY, USA. [42]University of Kentucky-Department of Pathology and Laboratory Medicine, University of Kentucky-Chandler Medical Center, Lexington, KY, USA. [43]University of Utah Huntsman Cancer Institute, Salt Lake City, UT, USA. [44]Department of Otolaryngology-Head and Neck Surgery, School of Medicine, Huntsman Cancer Institute, University of Utah, Salt Lake City, UT, USA. [45]Rutgers Cancer Institute, New Brunswick, NJ, USA. [46]Norris Comprehensive Cancer Center, Keck School of Medicine, University of Southern California, Los Angeles, CA, USA. [47]Department of Translational Genomics, Keck School of Medicine, University of Southern California, Los Angeles, CA, USA. [48]Murtha Cancer Center, Uniformed Services University/Walter Reed National Military Medical Center, Bethesda, MD, USA. [49]University of Oklahoma Stephenson Cancer Center, Oklahoma City, OK, USA. [50]Medical Oncology/Head and Neck Oncology, Stephenson Cancer Center, University of Oklahoma, Oklahoma City, OK, USA. [51]Indiana University School of Medicine, Indianapolis, IN, USA. [52]Roswell Park Comprehensive Cancer Center, Buffalo, NY, USA. [53]Departments of Surgical Oncology and Cancer Prevention and Control, Roswell Park Comprehensive Cancer Center, Buffalo, NY, USA. [54]University of Virginia Cancer Center, Charlottesville, VA, USA. [55]Department of Internal Medicine, Division of Hematology & Oncology, University of Virginia Health, Charlottesville, VA, USA. [56]Aster Insights, Hudson, FL, USA. [57]The Ohio State University Comprehensive Cancer Center and the Ohio State University College of Medicine, Columbus, OH, USA. [58]Department of Molecular Medicine and Therapeutics, The Ohio State University Comprehensive Cancer Center and the Ohio State University College of Medicine, Columbus, OH, USA. [59]Human Medical Genetics and Genomics Program, University of Colorado Anschutz, Aurora, CO, USA. [60]University of Colorado Cancer Center, University of Colorado Anschutz, Aurora, CO, USA. [61]Department of Epidemiology and Biostatistics, School of Public Health, Peking University, Beijing, China. [62]Peking University Center for Public Health and Epidemic Preparedness and Response, Beijing, China. [63]Key Laboratory of Epidemiology of Major Diseases (Peking University), Ministry of Education, Beijing, China. [64]Laboratory of Genome Technology, Human Genome Center, Institute of Medical Science, The University of Tokyo, Tokyo, Japan. [65]Laboratory of Clinical Genome Sequencing, Graduate School of Frontier Sciences, The University of Tokyo, Tokyo, Japan. [66]Research Service, Rocky Mountain Regional VA Medical Center, Aurora, CO, USA. [67]Transdivisional Research Program, Division of Cancer Epidemiology and Genetics, National Cancer Institute, National Institutes of Health, Bethesda, MD, USA. [68]Analytic and Translational Genetics Unit, Department of Medicine, Massachusetts General Hospital and Harvard Medical School, Boston, MA, USA. [69]University of Colorado Anschutz, Aurora, CO, USA. [70]Queen Mary University of London, London, UK. [71]The Institute of Medical Science, The University of Tokyo, Tokyo, Japan. [72]Department of Medicine, Harvard Medical School, Boston, MA, USA. *Lists of authors and their affiliations appear at the end of the paper. ✉e-mail: chris.gignoux@cuanschutz.edu; nikita.pozdeyev@cuanschutz.edu

**Penn Medicine BioBank**

Lindsay Guare[19] & Anurag Verma[19]

**Virtual Thyroid Biopsy Consortium**

Samantha L. White[1], Maizy S. Brasher[1], Jack Pattee[2], Wei Zhou[3,4,5], Sinéad Chapman[6], Yon Ho Jee[7], Jibril Hirbo[13,14], Nancy J. Cox[13,14], Peter Straub[13,14], Shinichi Namba[15,16,17], Emily Bertucci-Richter[18], Lindsay Guare[19], Ahmed Edris[20], Sam Morris[20], Ashley J. Mulford[21], Brian Fennessy[22], Martin D. Tobin[23,24], Jing Chen[23], Alexander T. Williams[23], Catherine John[23,24], David A. van Heel[25], Rohini Mathur[26], Sarah Finer[26], Marta R. Moksnes[27,28], Ben M. Brumpton[27,29,30], Bjørn Olav Åsvold[27,29,31], Raitis Peculis[32], Vita Rovite[32], Ilze Konrade[33], Ying Wang[3], Alan R. Sanders[21,35], David C. Whiteman[36], Stuart MacGregor[36], Sarah E. Medland[36], Unnur Thorsteinsdóttir[37,38], Kári Stefánsson[37,38], Robin G. Walters[20], Zhengming Chen[20], Liming Li[61,62,63], Koichi Matsuda[64,65], Yukinori Okada[15,16,17], Sebastian Zöllner[18], Anurag Verma[19], Michael H. Preuss[22], Eimear Kenny[22], Peter Kraft[7,67], Mark J. Daly[3,4,68], Benjamin M. Neale[3,4,68], Alicia R. Martin[3,4,68,72], Joanne B. Cole[1,59], Bryan R. Haugen[8,60], Christopher R. Gignoux[1,34,59,60] & Nikita Pozdeyev[1,8,34,60]

**Colorado Center for Personalized Medicine**

Kristy Crooks[34], Sameer Chavan[34], Matthew J. Fisher[34], Nicholas Rafaels[34], Jonathan A. Shortt[1,34], Milton Pividori[1,34,59], Christopher R. Gignoux[1,34,59,60] & Nikita Pozdeyev[1,8,34,60]

**Genes & Health Research Team**

David A. van Heel[25], Rohini Mathur[26] & Sarah Finer[26]

**The BioBank Japan Project**

Koichi Matsuda[64,65] & Yukinori Okada[15,16,17]

**Global Biobank Meta-analysis Initiative**

Wei Zhou[3,4,5], Sinéad Chapman[6], Mark J. Daly[3,4,68] & Benjamin M. Neale[3,4,68]

## Methods

### Ethical approval

The Colorado Multiple Institutional Review Board of the University of Colorado Denver Anschutz Medical Campus waived ethical approval for this work (COMIRB no. 20-2315). This study is the result of a large collaborative effort among multiple biobanks and programs. Cohort-specific GWAS analyses were performed by local researchers. Data collections for the cohorts were approved by local ethics committees. All biobank participants provided written informed consent. Participants in the biobanks were not compensated for their involvement in this study.

### VTB Consortium

We founded the VTB Consortium under the umbrella of the GBMI[23]. Nineteen biobanks from ten countries and four continents contributed GWAS results to the meta-analysis (Extended Data Fig. 1). Supplementary Table 1 lists the sizes of the biobank, ancestry strata, phenotyping, genotyping and imputation methods, and the software used for the GWAS.

### Phenotype definitions

We defined thyroid phenotypes using the International Classification of Diseases and Related Health Problems (ICD), Ninth (ICD-9-CM) and Tenth (ICD-10-CM) Revisions, Clinical Modifications billing codes for the United States biobanks, the ICD-9 and ICD-10 billing codes for international biobanks, and SNOMED codes and survey codes for the AoU (Supplementary Table 2). These phenotype definitions were shared with teams participating in the VTB Consortium.

To evaluate the performance of PRS and study their association with ThC risk phenotypes, we conducted clinical chart reviews for participants in the CCPM biobank. Histopathological and cytological diagnosis, patient characteristics (age at ThC diagnosis, death from ThC and risk of structural disease recurrence), tumor characteristics (tumor size, tumor focality, presence of extrathyroidal extension, lymphatic and angioinvasion, surgical margins positivity) and metastatic disease characteristics (presence of locoregional and distant metastases, extranodal extension, size and number of lymph node metastases) were extracted from surgical histopathology reports, thyroid nodule fine-needle aspiration reports and endocrinology notes.

The risk of structural disease recurrence was estimated on continuous (1–55% risk) and categorical scales as described in the ATA ThC guidelines[16]. For patients with multiple surgeries, the highest stage or risk was used (for example, if the first surgery's histopathology evaluation reported an Nx stage but lateral neck metastases were found later, the N1b stage was used for the association analysis). ThC annotations are listed in Supplementary Table 16. Benign cases for PRS evaluation in the CCPM cohort were defined based on surgical histopathology reports.

### GWAS

Case and control definitions for the GWAS are listed in Supplementary Table 3. Phenotype exclusions were used only if clinically or biologically justified. We excluded (1) patients diagnosed with medullary ThC from the ThC GWAS (if medullary ThC data were available because rare medullary ThCs are genetically distinct from common follicular cell-derived ThCs); (2) ThC cases from the BNG GWAS (because all ThCs are initially diagnosed as thyroid nodules to avoid contamination of BNG cases with malignant tumors), and (3) patients diagnosed with hypothyroidism other than primary (iatrogenic, congenital, central) from the hypothyroidism GWAS.

Each biobank conducted genotyping, imputation, quality control and genetic ancestry analysis independently (Supplementary Table 1) except for the AoU, which used a custom pipeline designed to leverage whole-genome sequencing data and maximize variant overlap with other biobanks.

GWAS analyses were run using either linear mixed models (SAIGE)[66] or whole-genome regression (REGENIE)[67], adjusted for case-control imbalances using saddlepoint approximation or Firth's logistic regression. The biobanks were instructed to use age, sex, up to 20 first principal components and biobank-specific variables, such as genotyping batches and recruiting centers, as covariates.

In addition to multi-ancestry analyses, GWAS stratified according to genetic ancestry were performed when the case counts permitted. Supplementary Table 4 lists case and control counts, Summix2 (ref. 29) population structure estimates and quality control metrics calculated with the cov-LDSC[28] for 198 GWAS.

### GWAS in the All of Us research program

We used All of Us whole-genome sequencing (WGS) v.7 data (245,388 WGS) to produce a genetic dataset that maximizes variant overlap with the analyses performed in the other biobanks (Extended Data Fig. 6 and 7). An inclusive list of single-nucleotide polymorphisms (SNPs) and indels from the GWAS analyses was compiled and supplemented with variants from the Polygenic Score Catalog (reported as of February 2024). This list contained ~147 million SNPs and indels.

WGS variant-level quality control was performed by All of Us, as outlined in the Research Program Genomic Research Data Quality Report[68]. In addition, we filtered the dataset to a maximal set of unrelated samples estimated from kinship scores and only included individuals with electronic health records or survey data for phenotype definitions (193,429 WGS).

We developed a Hail Python pipeline that extracts variants of interest from the All of Us variant dataset (https://hail.is/docs/0.2/vds/index.html). The code is publicly available in the GitHub repository (https://github.com/pozdeyevlab/vds-filter/tree/main). The resulting BGEN dataset contained ~118 million directly genotyped variants (a significant decrease from 972 million variants in the variant dataset), permitting GWAS.

### Post-GWAS quality control

The post-GWAS quality control workflow diagram is shown in Extended Data Fig. 8. All GWAS summary data were harmonized to gnomAD (v.4.1.0) (GRCh38 human genome reference)[30].

Each GWAS summary dataset (Supplementary Table 4) was processed using the following steps.

**Variant-level quality control.** The following variants were removed from the GWAS summary data: variants containing alleles with characters other than A, T, C or G; variants where a $P$ value could not be calculated (NA), effect size ($\beta$) or s.e. $\geq 1 \times 10^{-6}$ or $\leq -1 \times 10^{-6}$, and variants with an imputation score less than 0.3; variants with allele frequency less than 0.0005 or greater than 0.9995; and variants with allele count of less than 20. Variants were aligned to the gnomAD (v.4.1.0) reference 30. Ancestry-specific gnomAD allele frequencies were used for the single-ancestry GWAS. Both palindromic and non-palindromic variants were tested for exact and inverse alignments. Palindromic variants were removed because of potential strand flip if they met any of the following criteria: the fold difference between the gnomAD allele frequency and GWAS was greater than 2; or the GWAS allele frequency was greater than 0.4 and less than 0.6; or the GWAS allele frequency was less than less than 0.4 and the gnomAD allele frequency was greater than 0.6; or the GWAS allele frequency was greater than 0.4 and the gnomAD allele frequency was less than 0.6; variants flagged as low-quality by gnomAD; and variants with a Mahalanobis distance between the gnomAD allele frequency and a harmonized GWAS allele frequency of more than three s.d. from the mean.

**GWAS summary level data quality control.** Summix2 (ref. 29) was used to estimate the population structure from the GWAS summary data. We used a random set of 10,000 variants from chromosome

21 and reference allele frequencies for the AFR, AMR, EAS, NFE, MID and SAS genetic ancestry groups from gnomAD (v.4.1.0). The results from five Summix2 runs, each using a different random set of reference variants, were averaged. We compared GWAS-derived Summix2 population proportion estimates to those published by the Million Veteran Program[26], CCPM[69] and All of Us[68] and found near-perfect agreement (Extended Data Fig. 9, $r^2 = 0.999$, $P = 1.96 \times 10^{-22}$).

Single-ancestry GWAS summary data analysis showed good agreement between the ancestry reported by the biobank and the Summix2 estimate (fraction of target ancestry was 0.88–0.97).

Cov-LDSC[28] was used to evaluate confounding in the GWAS summary data, calculate the heritability of phenotypes and estimate the genetic correlation between thyroid diseases. For each major continental ancestry, we generated a custom reference panel of 5,000 WGS from All of Us. For the multi-ancestry GWAS, we used ancestry proportions calculated with Summix2 (Supplementary Table 4). Samples, regions and variants that met at least one of the following criteria were removed: (1) missingness of more than 0.1; (2) closely related individuals (plink king cutoff of 0.0884); (3) Hardy–Weinberg equilibrium exact test $P < 1 \times 10^{-6}$; (4) minor allele frequency of less than 0.01; and (4) genomic regions with high linkage disequilibrium (LD). Genetic principal components were calculated using plink2 (ref. 70). Ten principal components and a window of 20 cM were used to calculate covariate-adjusted LD scores and estimate the LD score regression intercept (Supplementary Table 4), heritability (Supplementary Table 7) and genetic correlations (Supplementary Table 8)[28].

## GWAS meta-analysis
A fixed inverse-variance-weighted meta-analysis was run using METAL[71]. Individual GWAS summary data with cov-LDSC $y$ axis intercepts significantly deviating from one were adjusted before the meta-analysis.

## Post-meta-analysis quality control, variant annotation and classification
To minimize the false positive hits introduced by confounding within a single large biobank, only variants present in at least four input GWAS datasets were considered in the downstream analysis. If three or fewer datasets were available for the ancestry-stratified meta-analysis, then the threshold was set to two. Cochran's $Q$ $P$ values were calculated to assess heterogeneity across datasets.

We used the hg38 human genome reference throughout the study. Phased $r^2$ values were computed using custom reference cohorts with matching population structure generated from the All of Us v.7 genomes ($n = 50,000$ for the mixed-ancestry, EUR-like and AFR-like meta-analyses; 40,000 and 5,500 for the AMR-like and EAS-like meta-analyses, respectively).

Genomic loci were defined using the LD clumping procedure implemented in PLINK 2.0 (ref. 70) with an index variant $P \leq 5 \times 10^{-8}$, 5-Mb search window and $r^2$ threshold of 0.01. Independent clumps were defined as those that did not share variants associated with the phenotype at $P \leq 1 \times 10^{-5}$. One variant with the lowest $P$ value from each independent clump was selected as a lead variant and reported in Supplementary Tables 6.1–6.6.

Lead variants were mapped to the nearest gene and annotated using ANNOVAR (version date 7 June 2020)[72]. A locus was considered new if no variants for the corresponding phenotype were reported within ±500 kb in the GWAS Catalog (as of April 2024; v.1.0.2)[31]. Otherwise, the variant was labeled as previously discovered.

## Heritability estimation and genetic correlation analysis
We used cov-LDSC (v.1.0.0)[28] with a custom-population-structure-matched LD reference panel to calculate SNP-based heritability ($h^2_{SNP}$). Observed-scale heritability estimates and the corresponding s.e. were converted to the liability scale using phenotype population prevalence

calculated in the All of Us v.7 dataset (Supplementary Table 7). Similarly, pairwise genetic correlations between the five thyroid phenotypes (Supplementary Table 8) were calculated using cov-LDSC with a custom LD score reference panel.

## PRS calculation and evaluation
To calculate and evaluate PRS, we performed a leave-CCPM-biobank-out GWAS meta-analysis. We also used a leave-CCPM-biobank-out GWAS meta-analysis from the GBMI phase I project[23] for comparison. All PRS in this study were tested on the out-of-sample CCPM dataset ($n = 94,651$). This approach minimizes inflation of PRS performance due to overfitting. Adjusted PRS (covariates of age, sex and ten genetic principal components) were cross-validated (fivefold).

The PRS was calculated as a weighted sum of independent genome-wide significant risk alleles. For the clinically relevant use case of distinguishing ThC from BNG, we defined $PRS_{ThC\ versus\ BNG}$ as the difference between the PRS for ThC ($PRS_{ThC\ versus\ All}$) and the PRS for BNG ($PRS_{BNG\ versus\ All}$): $PRS_{ThC\ versus\ BNG} = PRS_{ThC\ versus\ All} - PRS_{BNG\ versus\ All}$.

PRS performance predicting binary phenotypes was assessed using the AUC. AUCs were compared using the DeLong's test for significant differences.

## TWAS
We performed the $cis$-eQTL TWAS using FUSION[33]. FUSION was run on multi-ancestry and European meta-analysis summary data, 1,000 genomes LD reference data and all sample thyroid expression reference weights precomputed from GTEx v.8 (ref. 36) (http://gusevlab.org/projects/fusion/).

To replicate our findings in FUSION, we also used Summary-based PrediXcan (S-PrediXcan)[35] to derive gene-level association results from the GWAS summary statistics and GTEx v.8 (ref. 36) as the reference set. The GWAS meta-analysis summary data were harmonized and imputed as described previously (https://github.com/hakyimlab/summary-gwas-imputation). An imputed GWAS was used to generate gene-trait associations in thyroid gland tissue.

## Candidate gene expression and pathway analysis
We studied the gene expression of candidate genes linked to significant genetic associations using ANNOVAR annotation or the $cis$-eQTL TWAS (using a Bonferroni-corrected significance threshold). Intergenic variants that could not be attributed to the expressed gene were not included.

We compared mRNA expression of ThC-associated genes in 20 human tissues using the National Center for Biotechnology Information Gene database (www.ncbi.nlm.nih.gov/gene (ref. 38)). We investigated mRNA expression according to age at diagnosis, cancer stage, thyroid differentiation and other tumor features (Supplementary Table 10) in ThCs from the TCGA study[39] and the ORIEN AVATAR Program (www.oriencancer.org/research-programs). We accounted for common somatic oncogenic drivers using logistic (binary outcome; for example, presence of extrathyroidal extension), ordinal (for ordered categorical data; for example, disease stage) or linear (for continuous outcomes; for example, age at diagnosis) regression with index covariates for the presence of *BRAF* V600E or *H/N/KRAS* mutations.

The Reactome and KEGG pathway analyses were performed on all significant genes combined from FUSION and ANNOVAR using the ReactomePA (v.1.16.2)[46,47] and clusterProfiler[73] packages in $R$ v.4.4 with default Benjamini–Hochberg adjustment for multiple hypothesis testing.

## Statistics and reproducibility
For each significant locus, the number of biobanks with a significant association ($P < 5 \times 10^{-8}$) is listed in the Supplementary Tables 6.1–6.6. No statistical method was used to predetermine sample size. No data were excluded from the analysis. The experiments were not

randomized. The investigators were not blinded to allocation during the experiments and outcome assessment.

## Reporting summary

Further information on research design is available in the Nature Portfolio Reporting Summary linked to this article.

## Data availability

The GWAS meta-analysis summary data are available for download from the GWAS Catalog (www.ebi.ac.uk/gwas/; accession nos. GCST90627737–GCST90627776). The PRS weights have been deposited in the PGS Catalog (www.pgscatalog.org/; accession no. PGP000748; score IDs PGS005258-PGS005274). The proprietary, privately funded AVATAR data used in this study were generated by Aster Insights (www.asterinsights.com) and provided to support this project in collaboration with ORIEN. AVATAR data are not open source in public repositories; all inquiries regarding opportunities for data licensing (industry researchers) or collaboration with ORIEN (academic researchers) should be submitted to https://researchdatarequest.orienavatar.com. A follow-up with more information relevant to each specific inquiry is expected within five business days of submission.

## Code availability

Original code is publicly available from GitHub at https://github.com/pozdeyevlab/gwas-analysis and Zenodo https://doi.org/10.5281/zenodo.17468664 (ref. 74).

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

## Acknowledgements

We thank the participants and investigators from all participating biobanks. We thank Regeneron Genetics Center for genotyping part of the CCPM data. We gratefully acknowledge All of Us Research Program participants for their contributions, without whom this research would not have been possible. We also thank the All of Us Research Program of the National Institutes of Health (NIH) for making available the participant data examined in this study. We thank the Million Veteran Program participants and researchers who made the GWAS possible and results publicly available (dbGaP accession no. phs002453.v1.p1). We acknowledge ORIEN Members for their commitment to data sharing. The ThC gene expression data included in this work were obtained through the AVATAR Project. This research has been conducted using the UKB Resource under application no. 95339. Genes & Health thanks Social Action for Health, Centre of The Cell, members of our Community Advisory Group and staff who recruited and collected data from volunteers. We thank the National Institute for Health and Care Research (NIHR) National Biosample Centre (UK Biocentre), the Social Genetic & Developmental Psychiatry Centre (King's College London), the Wellcome Sanger Institute and the Broad Institute for sample processing, genotyping, sequencing and variant annotation. This work uses data provided by patients and collected by the National Health Service (NHS) as part of their care and support. This research used Queen Mary University of London's Apocrita HPC facility, supported by QMUL Research-IT (https://doi.org/10.5281/zenodo.438045)[75].

Genes & Health thanks Barts Health NHS Trust, the NHS Clinical Commissioning Groups (City and Hackney, Waltham Forest, Tower Hamlets, Newham, Redbridge, Havering, Barking and Dagenham), the East London NHS Foundation Trust, Bradford Teaching Hospitals NHS Foundation Trust, Public Health England (especially D. Wyllie), Discovery Data Service/Endeavour Health Charitable Trust (especially D. Stables), Voror Health Technologies Ltd (especially S. Don), NHS England (for what was NHS Digital) for GDPR-compliant data sharing backed by individual written informed consent.

The Trøndelag Health Study (HUNT) is a collaboration between the HUNT Research Centre (Faculty of Medicine and Health Sciences (M.H.), Norwegian University of Science and Technology (NTNU)), the Trøndelag County Council, the Central Norway Regional Health Authority and the Norwegian Institute of Public Health. The genotyping in HUNT was financed by the NIH, the Kristian Gerhard Jebsen Foundation, the University of Michigan, the Research Council of Norway, the Liaison Committee for Education, Research and Innovation in Central Norway and the Joint Research Committee between St Olav's Hospital and the Faculty of Medicine and Health Sciences at NTNU. The genotyping and imputation efforts in HUNT were a collaboration between researchers from the Department of Public Health and Nursing (I.S.M.; M.H., NTNU) and the University of Michigan Medical School and the University of Michigan School of Public Health. The genotyping was performed at the Genomics Core Facility (M.H., NTNU).

We thank the Michigan Genomics Initiative participants, AI & Digital Health Innovation at the University of Michigan, the University of Michigan Medical School Central Biorepository and the University of Michigan Advanced Genomics Core for providing data and specimen storage, management, processing and distribution services, and the Center for Statistical Genetics in the Department of Biostatistics at the School of Public Health for genotype data curation, imputation and management in support of the research reported in this publication. This project was funded by the National Cancer Institute (grant no. 1R21CA282380) to N.P., B.R.H. and C.R.G., and the Colorado Clinical and Translational Sciences Institute (CCTSI) grant CO-J-24-170 to N.P. CCTSI is supported in part by Colorado CTSA Grant UL1 TR002535 from NIH/NCATS.

The QSkin Study has been supported by grants from the Australian National Health and Medical Research Council (1185416, 1063061, 552429). We thank all the participants in the QSkin cohort.

L.F. is supported by the VA Merit Award no. I01BX006252, project no. MVP114. T.K. is supported by the Novo Nordisk Foundation Data Science Emerging Investigator grant (no. NNF20OC0062294). M.P. is supported by the National Human Genome Research Institute (no. R00HG011898). L.M.E. is supported by Hevolution/AFAR NI23013

and the National Institute of Aging (no. AG046938). S.M. and D.C.W. are supported by Australian National Health and Medical Research Council Investigator grants. P.K.'s research was supported in part by the Intramural Research Program of the NIH. The contributions of the NIH are considered Works of the United States Government. The findings and conclusions presented in this article are those of the authors and do not necessarily reflect the views of the NIH or the U.S. Department of Health and Human Services. W.Z. was supported by the National Human Genome Research Institute of the NIH under award no. K99/R00HG012222.

Genes & Health has recently been core-funded by Wellcome (WT102627, WT210561), the Medical Research Council (UK) (M009017, MR/X009777/1, MR/X009920/1), Higher Education Funding Council for England Catalyst, Barts Charity (845/1796), Health Data Research UK (for the London substantive site) and research delivery support from the NHS NIHR Clinical Research Network (North Thames). We acknowledge the support of the NIHR Research Barts Biomedical Research Centre (no. 203330), a delivery partnership of Barts Health NHS Trust, Queen Mary University of London, St George's University Hospitals NHS Foundation Trust and St George's University of London. Genes & Health has recently been funded by Alnylam Pharmaceuticals, Genomics and a Life Sciences Industry Consortium of AstraZeneca, Bristol Myers Squibb, GlaxoSmithKline Research and Development, Maze Therapeutics, Merck Sharp & Dohme, Novo Nordisk, Pfizer and Takeda Development Center Americas. This work was partially supported by the National Institute for Health Research (NIHR) Leicester Biomedical Research Centre (NIHR203327). The views expressed are those of the author(s) and not necessarily those of the National Health Service, the NIHR or the Department of Health and Social Care.

## Author contributions

N.P. and C.R.G. conceived, designed and supervised the project. S.L.W. and N.P. conducted the main analyses in this study. N.P. and S.L.W. wrote the initial draft of the paper. M.S.B. and J.B.C. contributed data and analyses from the UKB. J.H., N.J.C. and P.S. contributed data and analyses from the BioVU. S.N., K.M. and Y.O. contributed data and analyses from the BioBank Japan Project. E.B.-R. and S.Z. contributed data and analyses from the Michigan Genomics Initiative. L.G. and A.V. contributed data and analyses from the Penn Medicine BioBank. A.E., S. Morris, R.G.W., Z.C. and L.L. contributed data and analyses from the China Kadoorie Biobank. A.J.M. and A.R.S. contributed data and analyses from the Genomics Health Initiative. B.F., M.P. and E.K. contributed data and analyses from the Bio*Me*. J.P., S. Chavan, M.J.F., N.R., M.L., J.A.S. and K.C. contributed data and analyses from the biobank at the CCPM. Y.H.J., P.K. and A.R.M. contributed data and analyses from the Korean Cancer Prevention Study-II. M.D.T., J.C., A.T.W. and C.J. contributed data and analyses from the EXCEED study. D.A.v.H., R.M. and S.F. contributed data and analyses from the Genes and Health study. M.R.M., B.M.B. and B.O.Å. contributed data and analyses from the Trøndelag Health Study. R.P., V.R. and I.K. contributed data and analyses from the Latvian Biobank. Y.W. contributed data and analyses from the Mass General Brigham Biobank. D.C.W., S. MacGregor. and S.E.M. contributed data and analyses from the QSkin Sun and Health Study. U.T. and K.S. contributed data and analyses from deCODE Genetics. L.F. contributed data and analyses from the Million Veteran Program. K.M.E., T.B., H.C.M., G.R., B.S., C.S., M.D.P., J.L.F., S.E., V.K., M.L.C., R.J.R., P.L.B. and M.D.R. contributed the ORIEN AVATAR study data. C.H.A. and L.M.E. helped define independent genetic loci using the LD clumping procedure. H.Z. and M.P. performed the S-PrediXcan analyses. C.C.B., T.L.J., M.B., C.D.R. and B.R.H. collected and interpreted the ThC clinical data. T.K. contributed to the interpretation of the hypothyroidism meta-analysis data. R.S. assisted with the interpretation of the pathway analysis data. A.E.H. advised on the use of Summix2 for the population structure estimates. W.Z., S. Chapman, M.J.D. and B.M.N. led the Global Biobank Meta-analysis Initiative. All authors contributed to and approved the final version of the manuscript.

## Competing interests

N.P. and B.R.H. received research support from Veracyte, unrelated to this study. B.R.H. is the Clinical Liaison for ThyroSeq at Sonic Healthcare USA. N.P., S.L.W., B.R.H. and C.R.G. filed a provisional patent application 151077-00053PR2 with the United States Patent and Trademark Office dedicated to the use of a PRS for ThC diagnosis. The other authors declare no competing interests.

## Additional information

**Extended data** is available for this paper at https://doi.org/10.1038/s41588-025-02483-w.

**Correspondence and requests for materials** should be addressed to Christopher R. Gignoux or Nikita Pozdeyev.

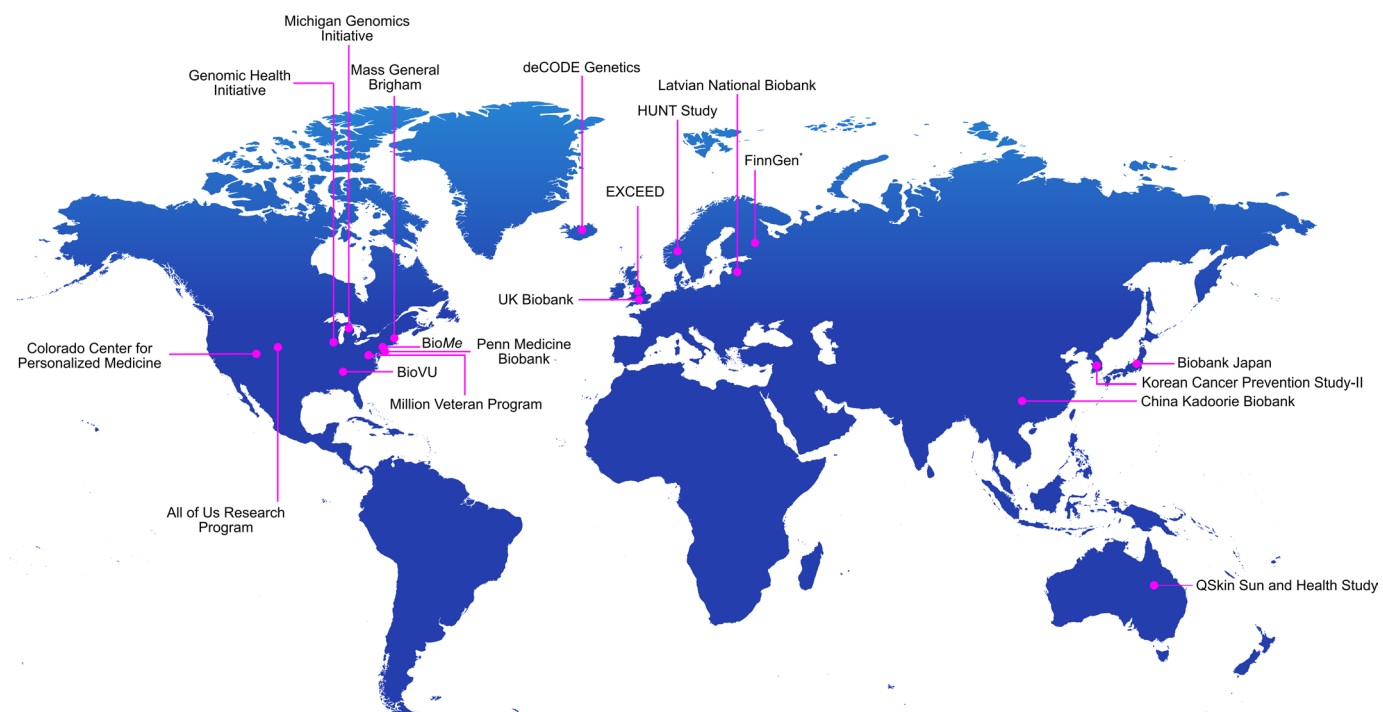

**Extended Data Fig. 1 | Virtual Thyroid Biopsy Consortium.** The Consortium aggregated data from 19 biobanks, 10 countries, four continents, and ~2.9 million participants.

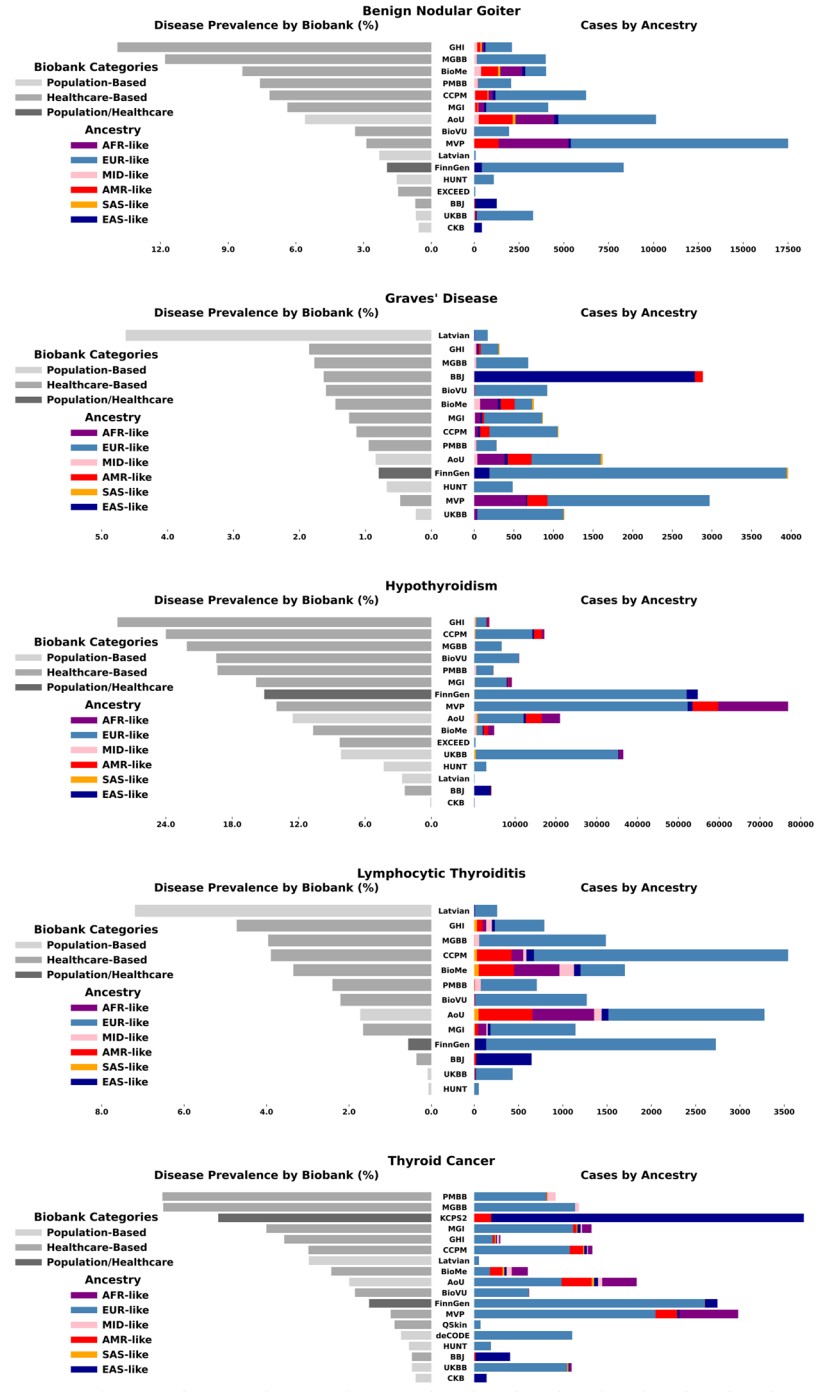

**Extended Data Fig. 2 | Prevalence of thyroid diseases across biobanks.** Genetic ancestry was estimated from GWAS summary data using Summix2.

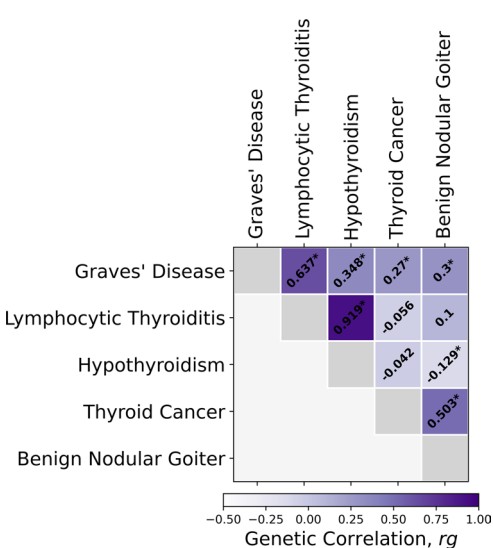

**Extended Data Fig. 3 | Genetic correlation analysis for thyroid diseases in EUR-like GWAS meta-analysis.** Genetic correlations were estimated using covariate-adjusted linkage disequilibrium score regression. The asterisks denote Benjamini-Hochberg false discovery rate (FDR) < 0.05; p-values were generated using a two-sided Wald test.

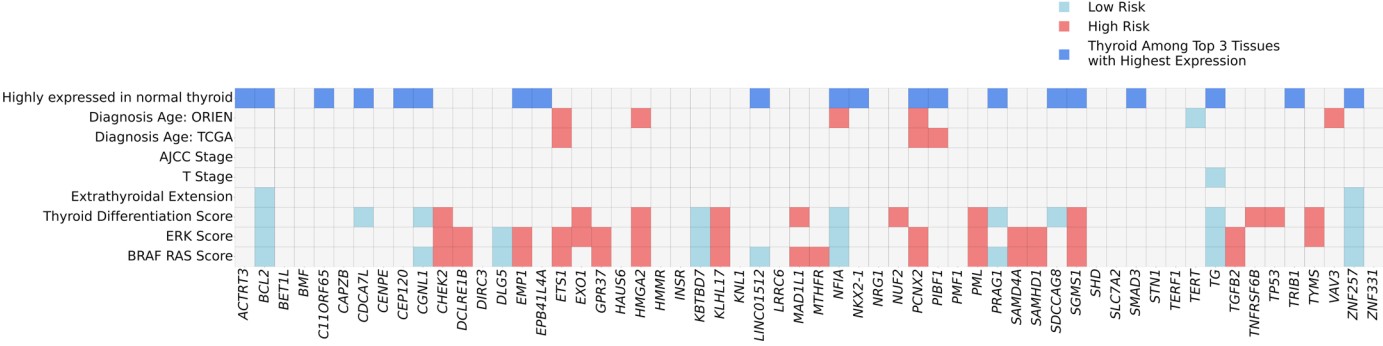

**Extended Data Fig. 4 | mRNA expression of thyroid cancer-associated genes in normal thyroid tissue and thyroid cancer.** Genes were identified from ANNOVAR annotations of genome-wide significant variants in thyroid cancer GWAS meta-analysis and FUSION TWAS cis-eQTL analysis. The dark blue color indicates genes with high expression in normal thyroid tissue, where the thyroid is among the top three tissues with the highest expression in pan-tissue transcriptome analysis from the NCBI Gene database (https://www.ncbi.nlm.nih.gov/gene). Positive (red) and negative (light blue) significant associations of mRNA expression with high-risk thyroid cancer features, such as earlier age at diagnosis, higher ERK score and lower thyroid differentiation score, etc., are shown. P-values were derived from a two-tailed t-test for linear regression (continuous variables) and a two-sided Wald test for logistic/ordinal regression (binary/ordinal variables). All regression analyses were adjusted for the major somatic oncogenic drivers, including *BRAF* V600E and *N/H/KRAS*. Significance threshold was adjusted using Bonferroni correction (p-value ≤ 1e-04). ORIEN - Oncology Research Information Exchange Network; TCGA – The Cancer Genome Atlas; AJCC - American Joint Committee on Cancer.

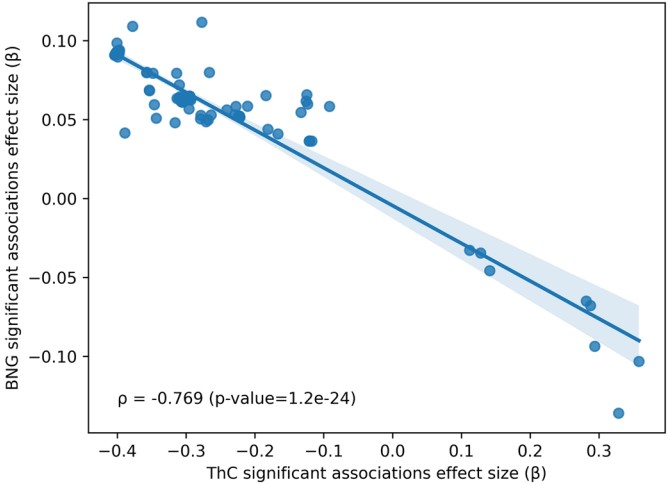

**Extended Data Fig. 5 | Scatterplot of effect sizes of the variants in *PTCSC2* locus significantly (p-value < 5e-8) associated with thyroid cancer and benign nodular goiter.** ThC – thyroid cancer. BNG – benign nodular goiter. $\rho$- Spearman correlation. Shading highlights the regression line's 95% confidence interval. P-value was calculated with a two-tailed t-test.

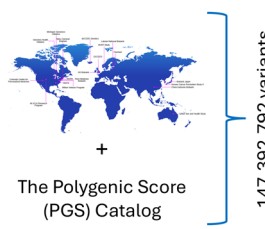

1. Split variant dataset object by chromosome
2. Split multiallelic sites
3. Subset selected variants
4. Densify genotypes
5. Annotate rows
6. Drop irrelevant fields
7. Remove variants with allele count <20
8. Add dosages
9. Add rsid information
10. Export in BGEN format

The Polygenic Score (PGS) Catalog

147,392,792 variants

**Extended Data Fig. 6 | All of Us Research Program whole genome sequencing data analysis pipeline.** Variants (SNPs and indels) from participating biobanks GWAS summary data and the Polygenic Score Catalog (https://www.pgscatalog.org) were extracted from the All of Us Research Program Hail variant dataset v7 object.

**Extended Data Fig. 7 | Variant overlap in GWAS from participating Biobanks.** A fraction of variants that are identical by chromosome, position, reference and alternate allele in the harmonized GWAS summary are shown. The All of Us Research Program GWAS (top row) was performed on whole-genome sequencing data and was designed to maximize variant overlap with other biobanks.

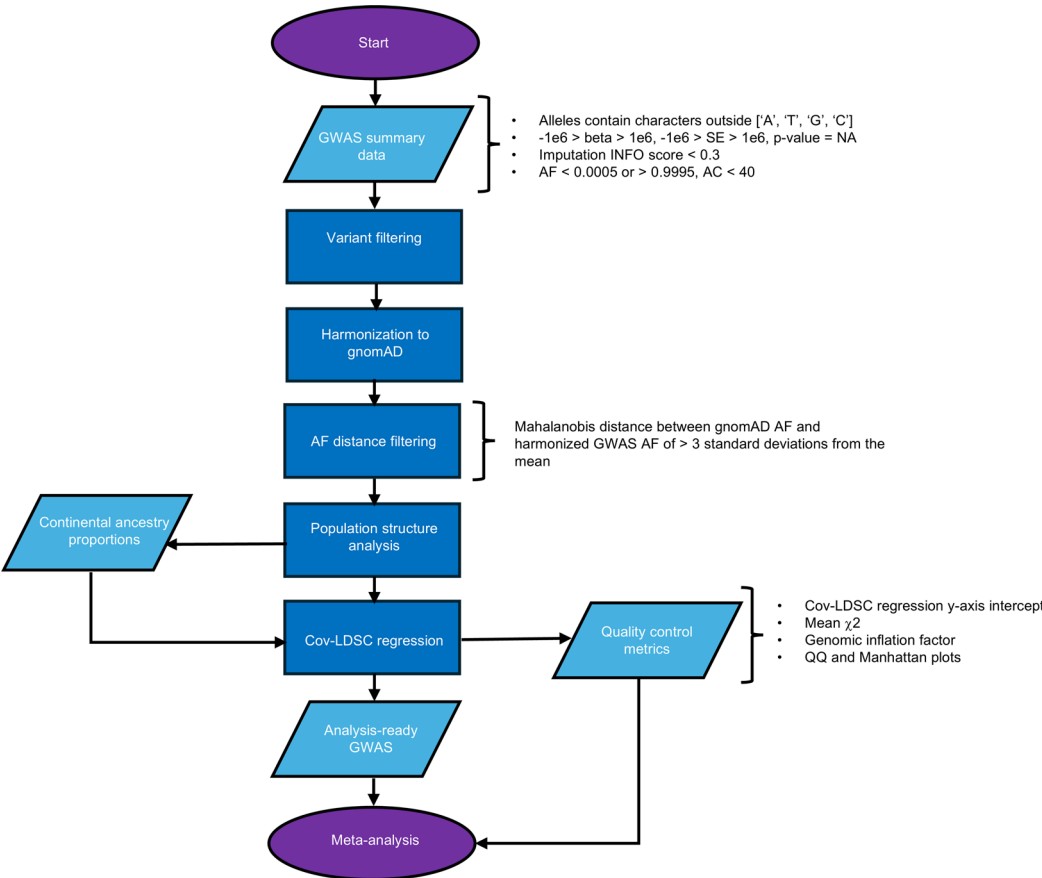

**Extended Data Fig. 8 | Post-GWAS quality control pipeline.** AF – allele frequency. AC – allele count, cov-LDSC – covariate-adjusted linkage disequilibrium score regression. QQ plot – quantile-quantile plot.

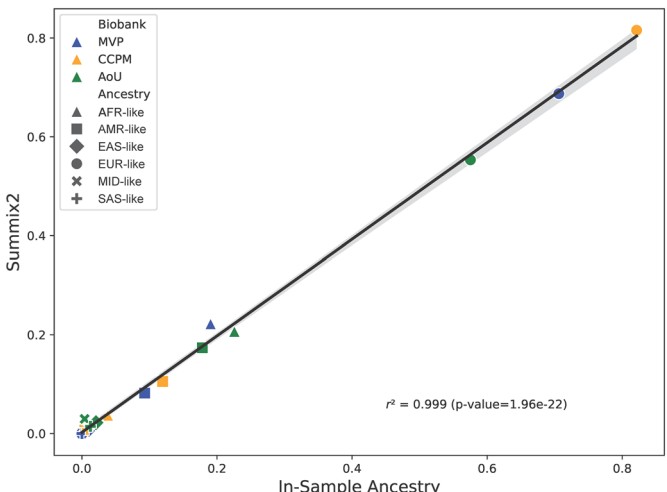

**Extended Data Fig. 9 | Correlation of major continental ancestry fractions estimated by Summix2 (y-axis) and published by the Million Veteran Program, Colorado Center for Personalized Medicine and All of Us Research Program Biobanks (x-axis).** Multi-ancestry GWAS summary data were used for this analysis. Shading highlights the regression line's 95% confidence interval. Pearson correlation coefficient p-value was calculated with a two-tailed t-test. MVP – Million Veteran Program. CCPM – Colorado Center for Personalized Medicine. AoU – All of Us Research Program.

# Reporting Summary

## Statistics

For all statistical analyses, confirm that the following items are present in the figure legend, table legend, main text, or Methods section.

| n/a | Confirmed | |
|---|---|---|
| ☐ | ☒ | The exact sample size ($n$) for each experimental group/condition, given as a discrete number and unit of measurement |
| ☒ | ☐ | A statement on whether measurements were taken from distinct samples or whether the same sample was measured repeatedly |
| ☐ | ☒ | The statistical test(s) used AND whether they are one- or two-sided *Only common tests should be described solely by name; describe more complex techniques in the Methods section.* |
| ☐ | ☒ | A description of all covariates tested |
| ☐ | ☒ | A description of any assumptions or corrections, such as tests of normality and adjustment for multiple comparisons |
| ☐ | ☒ | A full description of the statistical parameters including central tendency (e.g. means) or other basic estimates (e.g. regression coefficient) AND variation (e.g. standard deviation) or associated estimates of uncertainty (e.g. confidence intervals) |
| ☐ | ☒ | For null hypothesis testing, the test statistic (e.g. $F$, $t$, $r$) with confidence intervals, effect sizes, degrees of freedom and $P$ value noted *Give P values as exact values whenever suitable.* |
| ☒ | ☐ | For Bayesian analysis, information on the choice of priors and Markov chain Monte Carlo settings |
| ☒ | ☐ | For hierarchical and complex designs, identification of the appropriate level for tests and full reporting of outcomes |
| ☐ | ☒ | Estimates of effect sizes (e.g. Cohen's $d$, Pearson's $r$), indicating how they were calculated |

*Our web collection on statistics for biologists contains articles on many of the points above.*

## Software and code

Policy information about availability of computer code

| | |
|---|---|
| Data collection | Samples and genetic data were collected by participating biobanks. |
| Data analysis | Original code is publicly available at GitHub at https://github.com/pozdeyevlab/gwas-analysis and Zenodo  https://doi.org/10.5281/zenodo.17468664 74.<br><br>ANNOVAR version date June 7, 2020; https://annovar.openbioinformatics.org/en/latest/<br>Bcftools v1.16 ; https://samtools.github.io/bcftools/bcftools.html<br>LD score regression v1.0.1; https://github.com/bulik/ldsc<br>Covariate adjusted LD score regression, v1.0.0; https://github.com/immunogenomics/cov-ldsc/blob/master/ldsc.py<br>GNOMAD v4.0; https://gnomad.broadinstitute.org/data#v4<br>HAIL version v0.2.91; https://hail.is/docs/0.2/api.html<br>METAL version date May 5, 2020; https://genome.sph.umich.edu/wiki/METAL_Documentation<br>PLINK v1.9; https://www.cog-genomics.org/plink/<br>PLINK2 v2.0; https://www.cog-genomics.org/plink/2.0/<br>PRScs version date May 14, 2024; https://github.com/getian107/PRScs<br>REGENIE v3.2.4; https://rgcgithub.github.io/regenie/<br>S-PrediXcan v0.73; https://github.com/hakyimlab/MetaXcan<br>SAIGE v1.5.0.2; https://saigegit.github.io/SAIGE-doc/<br>Summix2 v2.8.0; https://www.bioconductor.org/packages/release/bioc/vignettes/Summix/inst/doc/Summix.html<br>TWAS/FUSION;  http://gusevlab.org/projects/fusion/<br>reactome.db v1.86.2; https://reactome.org/ |

For manuscripts utilizing custom algorithms or software that are central to the research but not yet described in published literature, software must be made available to editors and reviewers. We strongly encourage code deposition in a community repository (e.g. GitHub). See the Nature Portfolio guidelines for submitting code & software for further information.

ReactomePA v 1.46.0 ; https://pubmed.ncbi.nlm.nih.gov/26661513/
clusterProfiler v4.10.1; https://www.cell.com/the-innovation/fulltext/S2666-6758(21)00066-7?_returnURL&
R v4.3.1; https://www.r-project.org/
Python v3.11.10; https://www.python.org/downloads/

For manuscripts utilizing custom algorithms or software that are central to the research but not yet described in published literature, software must be made available to editors and reviewers. We strongly encourage code deposition in a community repository (e.g. GitHub). See the Nature Portfolio guidelines for submitting code & software for further information.

## Data

Policy information about availability of data

All manuscripts must include a data availability statement. This statement should provide the following information, where applicable:
- Accession codes, unique identifiers, or web links for publicly available datasets
- A description of any restrictions on data availability
- For clinical datasets or third party data, please ensure that the statement adheres to our policy

The GWAS meta-analysis summary data are available for download from the GWAS Catalog (https://www.ebi.ac.uk/gwas/; accession numbers GCST90627737-GCST90627776). The PRS weights are deposited in the PGS Catalog (https://www.pgscatalog.org/; accession number PGP000748).
The proprietary Avatar data used in this study was generated through private funding by Aster Insights (www.asterinsights.com) in collaboration with the Oncology Research Information Exchange Network (ORIEN®, www.oriencancer.org) and is not open source in public repositories; all inquiries regarding access to the data or collaboration within ORIEN should be submitted to the corresponding author or https://researchdatarequest.orienavatar.com/.
Further information and requests for resources should be directed to and will be fulfilled by the lead contact, Nikita Pozdeyev, email: nikita.pozdeyev@cuanschutz.edu.
Original code is publicly available at GitHub at https://github.com/pozdeyevlab/gwas-analysis and Zenodo  https://doi.org/10.5281/zenodo.17468664.

## Research involving human participants, their data, or biological material

Policy information about studies with human participants or human data. See also policy information about sex, gender (identity/presentation), and sexual orientation and race, ethnicity and racism.

| | |
|---|---|
| Reporting on sex and gender | Both sexes were included in the study and sex was used as a covariate in the analyses. Study findings apply to both sexes. Sex was determined from the genotyping data. Gender information was not collected or used in this study. |
| Reporting on race, ethnicity, or other socially relevant groupings | Genetically inferred ancestry was estimated as described in the Supplementary Table 1 for each participating biobank. Ancestry stratified meta-analyses were performed and results reported. Race or ethnicity was not used as a proxy for genetic ancestry. |
| Population characteristics | Population characteristics for each genome-wide association study are reported in the Supplementary Table 4. |
| Recruitment | Information on recruitment for each biobank is reported in the Supplementary Table 1. |
| Ethics oversight | Colorado Multiple Institutional Review Board for the University of Colorado Denver Anschutz Medical Campus, Aurora, Colorado, USA, waived ethical approval for this work (COMIRB #20-2315). This study is the result of a large collaborative effort among multiple biobanks and programs. Cohort-specific GWAS analyses were performed by local researchers. Data collections for the cohorts were approved by local ethics committees. Biobank participants were not compensated for their involvement in this study. |

Note that full information on the approval of the study protocol must also be provided in the manuscript.

# Field-specific reporting

Please select the one below that is the best fit for your research. If you are not sure, read the appropriate sections before making your selection.

☒ Life sciences          ☐ Behavioural & social sciences          ☐ Ecological, evolutionary & environmental sciences

For a reference copy of the document with all sections, see nature.com/documents/nr-reporting-summary-flat.pdf

# Life sciences study design

All studies must disclose on these points even when the disclosure is negative.

| | |
|---|---|
| Sample size | No predetermined sample size was used. All available data from participating biobanks was aggregated to maximize discovery power. |
| Data exclusions | No data were excluded from the analyses |
| Replication | Only variants tested by 4 or more biobanks were included in the multi-ancestry meta-analysis to ensure consistency across data sets. Heterogeneity was calculated using Cochran's Q measure. The number of biobanks with significant associations is reported for each variant. |

| | |
|---|---|
| Randomization | There is no allocation to experimental groups in this study. Randomization was not performed. |
| Blinding | Blinding is not relevant for this study. |

# Reporting for specific materials, systems and methods

We require information from authors about some types of materials, experimental systems and methods used in many studies. Here, indicate whether each material, system or method listed is relevant to your study. If you are not sure if a list item applies to your research, read the appropriate section before selecting a response.

## Materials & experimental systems

| n/a | Involved in the study |
|---|---|
| ☒ ☐ | Antibodies |
| ☒ ☐ | Eukaryotic cell lines |
| ☒ ☐ | Palaeontology and archaeology |
| ☒ ☐ | Animals and other organisms |
| ☒ ☐ | Clinical data |
| ☒ ☐ | Dual use research of concern |
| ☒ ☐ | Plants |

## Methods

| n/a | Involved in the study |
|---|---|
| ☒ ☐ | ChIP-seq |
| ☒ ☐ | Flow cytometry |
| ☒ ☐ | MRI-based neuroimaging |

## Plants

| | |
|---|---|
| Seed stocks | *Report on the source of all seed stocks or other plant material used. If applicable, state the seed stock centre and catalogue number. If plant specimens were collected from the field, describe the collection location, date and sampling procedures.* |
| Novel plant genotypes | *Describe the methods by which all novel plant genotypes were produced. This includes those generated by transgenic approaches, gene editing, chemical/radiation-based mutagenesis and hybridization. For transgenic lines, describe the transformation method, the number of independent lines analyzed and the generation upon which experiments were performed. For gene-edited lines, describe the editor used, the endogenous sequence targeted for editing, the targeting guide RNA sequence (if applicable) and how the editor was applied.* |
| Authentication | *Describe any authentication procedures for each seed stock used or novel genotype generated. Describe any experiments used to assess the effect of a mutation and, where applicable, how potential secondary effects (e.g. second site T-DNA insertions, mosiacism, off-target gene editing) were examined.* |

