## [Peer review file · Nature Genetics]

Global multi-ancestry genome-wide analyses identify genes and biological pathways associated with thyroid cancer and benign thyroid diseases

Corresponding Author: Dr Nikita Pozdeyev

This manuscript has been previously reviewed at another journal. This document only contains information relating to versions considered at Nature Genetics.

Version 0:

Decision Letter:

22nd Jul 2025

Dear Dr Pozdeyev,

Your Article, "Global multi-ancestry genetic study elucidates genes and biological pathways associated with thyroid cancer and benign thyroid diseases" has now been seen by 3 referees. You will see from their comments below that while they find your work of interest, some important points are raised. We are interested in the possibility of publishing your study in Nature Genetics, but would like to consider your response to these concerns in the form of a revised manuscript before we make a final decision on publication.

To guide the scope of the revisions, the editors discuss the referee reports in detail within the team, including with the chief editor, with a view to identifying key priorities that should be addressed in revision and sometimes overruling referee requests that are deemed beyond the scope of the current study. We hope that you will find the prioritized set of referee points to be useful when revising your study. Please do not hesitate to get in touch if you would like to discuss these issues further.

We therefore invite you to revise your manuscript taking into account all reviewer and editor comments. Please highlight all changes in the manuscript text file. At this stage we will need you to upload a copy of the manuscript in MS Word .docx or similar editable format.

*2) If you have not done so already please begin to revise your manuscript so that it conforms to our Article format instructions, available

[here](http://www.nature.com/ng/authors/article_types/index.html).

*3) Include a revised version of any required Reporting Summary: <https://www.nature.com/documents/nr-reporting-summary.pdf>

Please be aware of our [guidelines](https://www.nature.com/nature-research/editorial-policies/image-integrity) on digital image standards.

EXTENDED DATA FIGURES

Link Redacted

We hope to receive your revised manuscript within four to eight weeks. If you cannot send it within this time, please let us know.

Sincerely,

Safia Danovi, PhD
Senior Editor, Nature Genetics
ORCID: 0009-0007-7822-5479

Referee expertise:

Referee #1: GWAS, thyroid cancer

Referee #2: GWAS, thyroid diseases

Referee #3: GWAS, thyroid diseases

Reviewers' Comments:

Reviewer #1 (Remarks to the Author):

The authors use comprehensive genetic data from 19 biobanks to evaluate the genetic basis of thyroid diseases, including cancer, benign nodular goiter, thyroid autoimmune diseases (Graves' disease, lymphocytic thyroiditis/Hashimoto's), and primary hypothyroidism. Given the complex interrelationships among these conditions and the important clinical challenges of correctly diagnosing them, further investigation of their genetic basis is of great interest.

Overall, the analytic methods are sound, the data are well presented, and the manuscript is well written.

1. Figure 2 is a bit confusing because there are many colors and layers to the figure. Clarification of the multiple levels of rings would be useful, and indication of the meaning of the colors in the heatmap also would improve clarity. Ideally, I would prefer to see the actual correlation coefficients indicated, or at least a color scale like the one in Extended Data Fig. 3. Is there some other way to simplify Figure 2 overall?
2. Given that the oncogenic driver is a key determinant of gene expression and that driver also has a known strong relationship with other key characteristics (age at PTC, stage), can expression analyses that don't account for driver really be interpreted?

3. Why were independent loci of interest defined by distance (>500kB) rather than LD metrics?
4. It was confusing to follow the discovery versus test sets for construction of the PRS. Weren't all samples included in the multi-ancestry meta-analysis? If so, then none of them are truly independent for calculating the AUC.
5. Given the heterogeneity of thyroid cancer, can the authors comment on the histologic distribution?
6. How consistent and reliable is the distinction between lymphocytic thyroiditis primary hypothyroidism in the different studies?

Reviewer #2 (Remarks to the Author):

The authors conducted a GWAS meta-analysis for thyroid diseases including thyroid cancer, goiter, autoimmune and other primary hypothyroidism, and Graves' by including data of 19 Biobanks from 10 countries. Using the meta-analysis results, several additional in silico analyses like genetic correlation, pathways analyses, and transcriptome-wide association study (TWAS) were conducted. A focus was also put on polygenic risk score (PRS) calculation and its application on thyroid cancer diagnosis. The paper is generally well written, the topic and results are very interesting and relevant, and the large multi-ethnic sample represents a strength of the current study. However, I see several issues listed below that would need clarification. A major concern is related to the definition of the GWAS controls. Furthermore, I got the impression that the paper could be improved by applying more sophisticated methods on summarizing/identifying the GWAS hits and on summarizing the TWAS results.

1. From my point of view, there seem to be some issues related to the case-control definitions according to the information provided in Tables S2 and S3. E.g., autoimmune thyroiditis (ICD-10: E06.3) is intentionally not included in the case definition of hypothyroidism but seems to be included in the controls. Furthermore, the controls of the Graves' disease seem to include also individuals with other autoimmune thyroid diseases which may at least partially share the same genetic associations. Finally, the large number of individuals with primary hypothyroidism are included in controls of all other traits. I see at least the genetic correlation results and the assignment of known/novel loci (which were based on the trait label as appeared in the GWAS catalog) affected by this issue due to potential pleiotropic effects of loci associated among thyroid diseases.
2. The annotation of the hits to know/novel loci should be re-fined. First, the p-values of the GWAS catalog should be limited to $<5E-8$ which does not always be the case according to Tables S5.x. Furthermore, the corresponding trait names of the GWAS catalog should be provided. Do the lookup-traits really correspond to a comparable trait (and case/control) definition of the current study (which should be the case)? One randomly picked example of Table S5.5 (hypothyroidism) is rs11204752 labeled as known, but is reported (as of today) only as association for TSH, neuroblastoma, and tonsillitis.
3. The identification of independently associated loci (per trait) is somewhat outdated. After checking the stated reference to the GBMI flagship paper, it seems that the loci are simply combined using variants with the smallest association p-value and a distance window. Instead, state-of-the art methods that take the linkage disequilibrium (LD) structure into account should be used to identify independent associations (and subsequently loci if needed). This would be feasible as both overall and ancestry-specific reference panel are available in the current study for LD estimation.
4. Please check the numbers of cases and controls provided in Table S3 as there seems to be a mix-up with the trait labels.
5. Post-GWAS QC: Did the authors ensure that the p-value=0 filter did not remove associations where very small and p-values were (technically) provided by the GWAS software as zero, i.e. inadvertently removing highly significant true positive associations? For which reason the p=0 GWAS results were excluded, particularly since the inverse-variance weighted meta-analysis performed in METAL does not require p-values at all? How many variants were affected by this filter?
6. The AF harmonization filter of the palindromic variants requires some clarifications as currently written in line 1087. I assume that variants with fold change (not difference?) between gnomAD AF and GWAS AF > 2 are excluded. If this filter is applied regardless of the minor allele frequencies, I see a potential bias in removing low frequency variants due to their natural variation instead of potential strand flip.
7. The reported summary of the lead variant annotation is less informative without considering credible sets. In this case, small statistical/by chance variation in the association p-value may result in different lead variants with different functional annotations which do not necessarily reflect the underlying functional variant.
8. More information regarding the ancestry specific results per trait should be provided, i.e. which and how many variants were ancestry-specific or had different effect estimates across ancestry. Which are possible reasons that the TERF1 (8:73046129:G:A) variant was not significant in the mixed ancestry GWAS (e.g. monomorphic or due to different effect estimates in other ancestries)?
9. The TWAS results are also quite sparsely reported. Do significantly associated genes overlap between the traits (and which genes)? Are these genes overlap with results from the TSH and thyroxine TWAS in thyroid tissue (<https://doi.org/10.1530/ETJ-24-0067>)?
10. Taking into account that the study aimed distinguishing benign and malignant thyroid nodules using PRS, there should be information on the sensitivity and specificity provided, as well as calibration plots. The AUC alone provides only limited predictive value.

11. Please provide Manhattan and QQ plots of the GWAS meta-analysis results as Supplementary Figures.

12. The regional association plots in the Supplementary Figures should be colored accordingly to the LD information of the index variant with the surrounding associations.

13. I assume that there are limitations of the study, which should be stated in the discussion. On the other hand, please avoid statements like "largest GWAS meta-analysis" as this may be outdated quickly.

Minor:

- Introduction (line 541): I suggest to limit the references to peer-reviewed published papers, i.e. remove ref #22. Instead, the GWAS publication PMID: 38291025 could be referenced which includes (among other traits) also TSH and data from biobanks – although it might relativize the statement a bit that "systematic analysis of underlying genes, pathways, and clinical relevance is missing" for thyroid related traits.

- Please update the Summix2 reference to its final AJHG publication.

- Extended Data Figure 3: Please provide the numbers of the genetic correlations, e.g. in the lower part of the correlation matrix plot

Reviewer #3 (Remarks to the Author):

The current study comprises a large meta-analysis of the most important thyroid diseases using data from large biobanks. The major strengths include its sample size, and the analysis of thyroid diseases which are underrepresented in GWAS such as Graves and nodular disease. The methodology is solid and data are clearly presented. The identified pathways, especially for thyroid cancer and nodular disease provide a further insight into the pathogenesis of benign and malignant nodules. I have no concerns regarding all of these aspects and believe that this manuscript adds to the current literature. However, I do have the following concerns:

1. In the absence of in vitro or other in-depth follow-up analyses of the GWAS findings, one of the most attractive findings from a clinical perspective is the PRS for benign vs malignant nodules. The authors show that the extreme PRS percentiles are associated with an increased risk of malignant nodules. However, this finding is expected and the real clinical question is whether such a PRS is of additional value to current clinical practice, as it could very well be that it has no real additional value. E.g., how does this PRS compare to the choices made on ultrasound findings (TIRADS scores) and subsequent FNA findings. All of these data should be present in the charts and it would provide much more clarity on the use of these PRS.

2. The same holds true for the PRS on aggressiveness of thyroid cancer: where would this fit into clinical practice? The authors for example include lymph node metastases, but a neck ultrasound is part of the preoperative workup which is sensitive for detecting lymph node metastases. And if it would be used to detect cancers with higher risk of distant metastases, how does the PRS compare to other conventional tumor characteristics such as pathology characteristics and the presence of lymph node metastases. All of these data should also be present in the charts and it would provide much more clarity on the use of these PRS.

Version 1:

Decision Letter:

Our ref: NG-A69242R

22nd Sep 2025

Dear Dr Pozdeyev,

Thank you for submitting your revised manuscript "Global multi-ancestry genetic study elucidates genes and biological pathways associated with thyroid cancer and benign thyroid diseases" (NG-A69242R). It has now been seen by the original referees and their comments are below. The reviewers find that the paper has improved in revision, and therefore we'll be happy in principle to publish it in Nature Genetics, pending minor revisions to satisfy the referees' final requests and to comply with our editorial and formatting guidelines.

If the current version of your manuscript is in a PDF format, please email us a copy of the file in an editable format (Microsoft Word or LaTeX) – we can not proceed with PDFs at this stage.

Sincerely,

Safia Danovi, PhD
Senior Editor, Nature Genetics
ORCID: 0009-0007-7822-5479

Reviewer #2 (Remarks to the Author):

I thank the authors for addressing my questions satisfactorily including the justification of the applied case-control definition, and for conducting the additional analyses. There are no additional comments or concerns from my side, except that ref 65 in the Methods should also be replaced by 29 (Summix2).

Reviewer #3 (Remarks to the Author):

I thank the authors for this revised version of the manuscript. Please find below my comments:

Response to my first question: I am satisfied with the authors' response, but I would advise them to include these considerations shortly in the discussion, as I believe that these considerations regarding the place of this PRS in clinical practice will be of interest to the clinical readership.

Response to my second question: I am happy to see that the extra analyses have shown that this PRS could be of significant use in population screening for aggressive thyroid cancer, which is a real addition to this manuscript. However, my original question remains on the exact place of this PRS in the workup of a diagnosed thyroid cancer patient: The authors for example include lymph node metastases, but a neck ultrasound is part of the preoperative workup which is sensitive for detecting lymph node metastases. And if it would be used to detect cancers with higher risk of distant metastases, how does the PRS compare to other conventional tumor characteristics such as pathology characteristics and the presence of lymph node metastases? In other words: what is the real addition to clinical practice? In case this remains uncertain, this should be explicitly mentioned in the discussion (while acknowledging the promising findings reported in this manuscript).

Dear Dr. Danovi and Referees,

We are grateful for the opportunity to respond to the Referees' critiques and suggestions and submit a revised version of the manuscript. The study and the manuscript have improved thanks to the Referees' suggestions. Particularly high impact were the redesign of the algorithm for independent loci determination using a custom LD reference (Reviewers 1 and 2) and testing of PRS for aggressive thyroid cancer screening (Reviewer 3).

Please refer to our point-by-point responses, which are highlighted in blue.

Nikita Pozdeyev, on behalf of the team of authors.

Reviewers' Comments:

Reviewer #1 (Remarks to the Author):

The authors use comprehensive genetic data from 19 biobanks to evaluate the genetic basis of thyroid diseases, including cancer, benign nodular goiter, thyroid autoimmune diseases (Graves' disease, lymphocytic thyroiditis/Hashimoto's), and primary hypothyroidism. Given the complex interrelationships among these conditions and the important clinical challenges of correctly diagnosing them, further investigation of their genetic basis is of great interest.

Overall, the analytic methods are sound, the data are well presented, and the manuscript is well written.

1. Figure 2 is a bit confusing because there are many colors and layers to the figure. Clarification of the multiple levels of rings would be useful, and indication of the meaning of the colors in the heatmap also would improve clarity. Ideally, I would prefer to see the actual correlation coefficients indicated, or at least a color scale like the one in Extended Data Fig. 3. Is there some other way to simplify Figure 2 overall?

We significantly reworked Figure 2 to improve clarity. We adjusted colors (compatible with the most common type of color blindness), added the legend to the middle of each circular plot, and included correlation coefficients in the heatmap. We added variants loci attributed to protein-coding genes from ancestry-stratified meta-analyses, but did not show intergenic loci. All independent loci are listed in Supplementary Table 6.1-6.6. Genes in the "Nodular Diseases" plot correspond to the annotations in Supplementary Table 11.

We learned this method of illustrating genetic correlations, unique and pleiotropic associations, from other recent Nature publications studying complex interactions between phenotypes (e.g., PMID: 39653778). This way of presenting the data is superior to traditional Manhattan and Miami plots, and it is more information-rich. We agree with the reviewer that it requires extra cognitive effort.

2. Given that the oncogenic driver is a key determinant of gene expression and that driver also has a known strong relationship with other key characteristics (age at PTC, stage), can expression analyses that don't account for driver really be interpreted?

We are thankful for this suggestion. We updated gene expression analysis to account for major oncogenic drivers (Extended Data Figure 4, Supplementary Table 10). We now test for the association of gene mRNA expression with clinical features using logistic (binary outcome; e.g., presence of extrathyroidal extension), ordinal (for ordered categorical data; e.g., disease stage), or linear (for continuous outcomes; e.g., age at diagnosis) regression with index covariates for the presence of *BRAF* V600E or *H/N/KRAS* oncogenic drivers.

After adjustment, the expression of two more genes was significantly associated with younger age at diagnosis. None of the genes were associated with AJCC Stage. Fewer genes were associated with ERK score and BRAF/RAS score as expected.

3. Why were independent loci of interest defined by distance (>500kB) rather than LD metrics?

Following suggestions from Reviewers 1 and 2, we redefined independent loci using custom-calculated LD metrics. We assembled a reference cohort with matching population structure using the All of Us version 7 genotypes for each meta-analysis. We used these custom references to identify lead variants with $r^2 < 0.01$ within the 5 Mb LD window. Finally, we excluded clumps that share one or more variants with a GWAS p-value of $\leq 1e-5$ (suggestive significance threshold) with another clump. We generated locus plots with LD structure to visually confirm the independence of selected lead variants (Supplementary Figures 2 and 3).

We noted three improvements from using the LD-based strategy of defining independent loci: 1) we now excluded some loci previously considered independent, which are, in fact, in weak distant LD; 2) we identified novel independent loci (e.g., missense SNPs in *TG*); 3) PRS calculated from LD-based independent loci showed mild but consistent AUC increase.

4. It was confusing to follow the discovery versus test sets for construction of the PRS. Weren't all samples included in the multi-ancestry meta-analysis? If so, then none of them are truly independent for calculating the AUC.

We ran meta-analysis twice. For discovery and to maximize statistical power, we included data from all biobanks. Then, we repeated meta-analysis excluding CCPM GWAS and used these leave-CCPM-out summary data for PRS calculation and AUC estimates. For PRS evaluations, our training and validation samples are independent.

5. Given the heterogeneity of thyroid cancer, can the authors comment on the histologic distribution?

We have this information for the CCPM Biobank (our home cohort), where we performed a manual chart review of all patients with thyroid cancer billing codes. We added this data to the manuscript.

“Papillary thyroid cancer was the most common thyroid malignancy in the CCPM cohort (n=1024), followed by follicular thyroid carcinoma (n=41), oncocytic thyroid carcinoma (n=11), anaplastic thyroid carcinoma (n=7), and poorly differentiated thyroid carcinoma (n=4). For 257 thyroid cancer patients, histological subtype was not documented in the clinical records.”

Unfortunately, we do not have access to clinical charts from other Consortium sites, and thyroid cancer ICD-9-CM and ICD-10-CM codes do not differentiate thyroid cancer subtypes. We requested that medullary thyroid cancer patients be excluded from the analysis, where such data were available.

6. How consistent and reliable is the distinction between lymphocytic thyroiditis primary hypothyroidism in the different studies?

The billing codes for two conditions are different, and, of course, lymphocytic thyroiditis is the most common cause of hypothyroidism. Lymphocytic thyroiditis can be seen as a milder form of thyroid dysfunction when compared to primary hypothyroidism. We used lymphocytic thyroiditis as an internal control, testing the quality of phenotyping. We were satisfied with the near-perfect genetic correlation between lymphocytic thyroiditis and primary hypothyroidism despite different billing codes used. We do not make mechanistic conclusions based on the differences between these two phenotypes.

Reviewer #2 (Remarks to the Author):

The authors conducted a GWAS meta-analysis for thyroid diseases including thyroid cancer, goiter, autoimmune and other primary hypothyroidism, and Graves' by including data of 19 Biobanks from 10 countries. Using the meta-analysis results, several additional in silico analyses like genetic correlation, pathways analyses, and transcriptome-wide association study (TWAS) were conducted. A focus was also put on polygenic risk score (PRS) calculation and its application on thyroid cancer diagnosis. The paper is generally

well written, the topic and results are very interesting and relevant, and the large multi-ethnic sample represents a strength of the current study. However, I see several issues listed below that would need clarification. A major concern is related to the definition of the GWAS controls. Furthermore, I got the impression that the paper could be improved by applying more sophisticated methods on summarizing/identifying the GWAS hits and on summarizing the TWAS results.

1. From my point of view, there seem to be some issues related to the case-control definitions according to the information provided in Tables S2 and S3. E.g., autoimmune thyroiditis (ICD-10: E06.3) is intentionally not included in the case definition of hypothyroidism but seems to be included in the controls.

In this study, we focused on clinical primary hypothyroidism. Lymphocytic thyroiditis is the most common cause of primary hypothyroidism, and only 20-30% of patients with autoimmune thyroiditis will develop clinically significant disease (PMID: 31812326). Because of limited clinical relevance (with some exceptions, such as in pregnant women), autoimmune thyroiditis is infrequently diagnosed and coded by clinicians (explaining significantly smaller power of lymphocytic thyroiditis GWAS meta-analysis in our study). For these reasons, we chose not to include autoimmune thyroiditis (unless already diagnosed with hypothyroidism) in our cases for hypothyroidism GWAS. Instead, we run a separate, smaller meta-analysis dedicated to autoimmune thyroiditis as an internal control. Near-perfect genetic correlation between lymphocytic thyroiditis and hypothyroidism supports the high quality of our phenotype definitions.

Furthermore, the controls of the Graves' disease seem to include also individuals with other autoimmune thyroid diseases which may at least partially share the same genetic associations.

It is our belief that using control exclusions (unless clinically or physiologically justified, as in the case of secondary hypothyroidism in our GWAS#6) in large GWAS meta-analyses risks introducing bias that may lead to the discovery of spurious associations. We agree with the reviewer that patients with Graves' disease are more prone to developing other autoimmune diseases. Excluding all autoimmune diseases from controls but keeping them among the cases (in patients who also have Graves' disease) may lead to the discovery of autoimmune disease variants that are not associated with Graves' disease.

Finally, the large number of individuals with primary hypothyroidism are included in controls of all other traits. I see at least the genetic correlation results and the assignment of known/novel loci (which were based on the trait label as appeared in the GWAS catalog) affected by this issue due to potential pleiotropic effects of loci associated among thyroid diseases.

Please see our justification for not using control exclusions in response to the previous comment. By design, we wanted our cases and control definitions to differ only by the disease of interest. When the use of exclusion was unavoidable (e.g., removing thyroid cancer from benign goiter GWAS), we applied exclusions to both cases and controls to minimize bias. Excluding hypothyroidism from cases but not controls (and ~ 10% of individuals with other phenotypes also have primary hypothyroidism by chance) will lead to the discovery of hypothyroid associations even in GWAS for other thyroid diseases.

2. The annotation of the hits to know/novel loci should be re-fined. First, the p-values of the GWAS catalog should be limited to $<5E-8$ which does not always be the case according to Tables S5.x. Furthermore, the corresponding trait names of the GWAS catalog should be provided. Do the lookup-traits really correspond to a comparable trait (and case/control) definition of the current study (which should be the case)? One randomly picked example of Table S5.5 (hypothyroidism) is rs11204752 labeled as known, but is reported (as of today) only as association for TSH, neuroblastoma, and tonsillitis.

Thank you for pointing out these discrepancies. We refined our GWAS catalog comparison so that catalog variants with p-values greater than $5e-8$ are no longer included (Tables S6.1-6.6 column 'CATALOG_PVALUE'). Additionally, we refined our GWAS catalog comparisons to ensure that only relevant traits are included (column 'DISEASE/TRAIT' in Tables S6.1-6.6).

3. The identification of independently associated loci (per trait) is somewhat outdated. After checking the stated reference to the GBMI flagship paper, it seems that the loci are simply combined using variants with the smallest association p-value and a distance window. Instead, state-of-the art methods that take the linkage disequilibrium (LD) structure into account should be used to identify independent associations (and subsequently loci if needed). This would be feasible as both overall and ancestry-specific reference panel are available in the current study for LD estimation.

We addressed this critique as suggested by the Reviewer. Please see our response to Reviewer 1, comment # 3. We noted several improvements from this more sophisticated method of defining independent variants.

4. Please check the numbers of cases and controls provided in Table S3 as there seems to be a mix-up with the trait labels.

We apologize for this error, which is now corrected in a revised version of Supplementary Table 3.

5. Post-GWAS QC: Did the authors ensure that the p-value=0 filter did not remove associations where very small and p-values were (technically) provided by the GWAS

software as zero, i.e. inadvertently removing highly significant true positive associations? For which reason the $p=0$ GWAS results were excluded, particularly since the inverse-variance weighted meta-analysis performed in METAL does not require p-values at all? How many variants were affected by this filter?

This filter removed variants where the p-value could not be calculated (NA). This happens to a tiny fraction of variants (e.g., 114 out of 93,389,889 variants in the UK Biobank mixed ancestry hypothyroidism GWAS) when the model fails to converge. We changed the description of the filter accordingly.

The meta-analyzed results from METAL may output p-values that, if not handled properly, will be rounded to 0. We were aware of this and took steps to ensure that the data types used are compatible with such small numbers. For example, our smallest reported p-value of $6.29e-953$ (Supplementary Table 5.5, 9:97784318:C:T) is less than the smallest number the base R can handle ($5e-324$).

6. The AF harmonization filter of the palindromic variants requires some clarifications as currently written in line 1087. I assume that variants with fold change (not difference?) between gnomAD AF and GWAS AF > 2 are excluded. If this filter is applied regardless of the minor allele frequencies, I see a potential bias in removing low frequency variants due to their natural variation instead of potential strand flip.

We agree with the reviewer that the AF filter removes some of the rare palindromic variants that show discrepancies from gnomAD allele frequency due to chance/natural variation or errors in genotyping, imputation or variant calling. This could be especially true for smaller biobanks where rare variant AF calculations are less accurate.

For large biobanks such as All of Us and UK Biobank, AF estimates are reasonably concordant with gnomAD (in ancestry-matched reference) even for rare variants. We estimated that this filter removes $\sim 1.1\%$ of variants. In the setting of a very large GWAS in our study (up to 100 million variants), the negative effect from this filter is exceedingly small. In addition, this filter removes some truly erroneous variants similar to the Mahalanobis distance filter in the QC pipeline.

As we move from microarray genotyping data to WGS, the risk of erroneous rare genotypes will decrease, and we will test relaxing this filter in future analyses. We appreciate this thoughtful comment by the reviewer.

7. The reported summary of the lead variant annotation is less informative without considering credible sets. In this case, small statistical/by chance variation in the association p-value may result in different lead variants with different functional annotations which do not necessarily reflect the underlying functional variant.

We agree that we do not know if the lead variants are causal, and most are probably not. Fine mapping is on our 'to-do' list, although it is challenging for the heterogeneous mixed-ancestry meta-analysis (PMID: 36643910). The primary purpose of the table is 1 to illustrate that this meta-analysis led to the discovery of many novel loci, to show counts of exonic variants (likely to be causal), and, most importantly, to demonstrate how many loci could be attributed to the gene (other than intergenic variants). This information is important because our functional inference analyses (mRNA expression, pathway analysis and genetic susceptibility to thyroid cancer/BNG) are done at the level of the gene.

Acknowledging the Reviewer's critique, we moved Table 1 from the main text to Supplementary Tables.

8. More information regarding the ancestry specific results per trait should be provided, i.e. which and how many variants were ancestry-specific or had different effect estimates across ancestry. Which are possible reasons that the *TERF1* (8:73046129:G:A) variant was not significant in the mixed ancestry GWAS (e.g. monomorphic or due to different effect estimates in other ancestries)?

TERF1 (8:73046129:G:A) is a rare variant with an overall gnomAD allele frequency (AF) of 0.0006. The AF in non-Finnish Europeans is 0.00085, but the allele is ultra-rare in other ancestries (e.g., AF 0.00009529 in AFR-like population). Because of this difference, in many multi-ethnic cohorts, this variant AF was below the QC AF threshold of 0.0005, and it was removed prior to GWAS meta-analysis. The resulting statistical power for this variant in the EUR-like cohort was higher (N=1,303,270) than in the multi-ancestry cohort (n=809,661), explaining why it was discovered in the EUR-like meta-analysis only. This case illustrates the importance of ancestry-stratified analyses, as we performed in our study.

We made the following changes to the manuscript to highlight the value of ancestry-stratified analyses.

1. Ancestry-specific associations are now labeled with asterisks in Supplementary Tables 6.1-6.6.
2. We provide a total count of ancestry-specific loci (n=148).
3. We provide two mechanistically plausible examples of ancestry-specific associations: *TERF1* in thyroid cancer and *DIO1* in hypothyroidism.
4. Complete ancestry-stratified GWAS meta-analysis summary data files will become available for download from the NHGRI-EBI GWAS Catalog after manuscript acceptance for publication.

9. The TWAS results are also quite sparsely reported. Do significantly associated genes overlap between the traits (and which genes)? Are these genes overlap with results from the TSH and thyroxine TWAS in thyroid tissue (<https://doi.org/10.1530/ETJ-24-0067>)?

We performed these additional analyses (Supplementary Tables 9.3 and 9.4). The results were as expected.

“Consistent with a significant genetic overlap between thyroid diseases (Figure 2, Extended Figure 3, Supplementary Table 8), we found that many genes were discovered in more than one TWAS analysis (Supplementary Table 9.3). For example, cis-eQTLs and expression of *TGFB* were found to be associated with all thyroid diseases in our analysis and the TSH trait³⁷. Plausibly, most overlap in TWAS analyses was found between autoimmune thyroid diseases and TSH³⁷, a hormone that is clinically measured to diagnose hypothyroidism and Graves’ disease (Supplementary Table 9.4).”

10. Taking into account that the study aimed distinguishing benign and malignant thyroid nodules using PRS, there should be information on the sensitivity and specificity provided, as well as calibration plots. The AUC alone provides only limited predictive value.

The ThC vs.BNG PRS is not designed to be used alone, but as a component of a diagnostic test incorporating more than one risk assessment (for example, PRS plus ultrasound-image-based risk assessment, as in our publication [PMID: 37683082]). This is because even the best PRS only accounts for genetic risk (ignoring the environmental component) and will not achieve sufficient performance if used alone. Confusion matrix metrics for PRS alone are not particularly informative for this clinical use case.

In response to Reviewer 3's critique #2, we tested PRS as a screening test for aggressive thyroid cancer, a novel, even more exciting clinical application. In this case, PRS is intended to be used alone; therefore, we provided sensitivity and specificity estimates for a PRS threshold that will identify 80% of individuals with aggressive thyroid cancer in a healthcare system biobank (sensitivity: 0.803 [0.803; 0.803] and specificity: 0.569 [0.565; 0.572]).

We also note that in this study, we do not aspire to deliver a PRS that is ready for clinical use, but rather to inform which clinical applications for PRS are feasible. We agree that before clinical implementation, we will need to test calibrated PRS performance across sexes and ages prospectively in a clinical trial. This is a challenging but worthy objective that is out of scope for this project. We acknowledge this in the limitations section.

11. Please provide Manhattan and QQ plots of the GWAS meta-analysis results as Supplementary Figures.

We now provide Manhattan plots and QQ-plots for all mixed ancestry and ancestry-stratified meta-analysis (Supplementary Figure 1).

12. The regional association plots in the Supplementary Figures should be colored accordingly to the LD information of the index variant with the surrounding associations.

We recreated locus plots, including LD information from custom ancestry-matched references (Supplementary Figures 2 and 3).

13. I assume that there are limitations of the study, which should be stated in the discussion. On the other hand, please avoid statements like “largest GWAS meta-analysis” as this may be outdated quickly.

The major limitation of our study is the relatively low power to study non-EUR individuals, as we stated in the original submission. We now discuss other limitations as well.

We no longer describe our meta-analysis as “largest”.

Minor:

- Introduction (line 541): I suggest to limit the references to peer-reviewed published papers, i.e. remove ref #22. Instead, the GWAS publication PMID: 38291025 could be referenced which includes (among other traits) also TSH and data from biobanks – although it might relativize the statement a bit that “systematic analysis of underlying genes, pathways, and clinical relevance is missing” for thyroid related traits.

We no longer reference the preprint by Reeves et al. and cite PMID: 38291025 as suggested by the Reviewer.

- Please update the Summix2 reference to its final AJHG publication.

We now reference the final AJHG publication for all references to Summix2 in the manuscript.

- Extended Data Figure 3: Please provide the numbers of the genetic correlations, e.g. in the lower part of the correlation matrix plot

We have amended Figure 2 and Extended Data Figure 3 to show genetic correlations in a heatmap. Full statistical data is also available in Supplementary Table 8.

Reviewer #3 (Remarks to the Author):

The current study comprises a large meta-analysis of the most important thyroid diseases using data from large biobanks. The major strengths include its sample size, and the analysis of thyroid diseases which are underrepresented in GWAS such as Graves and nodular disease. The methodology is solid and data are clearly presented. The identified pathways, especially for thyroid cancer and nodular disease provide a further insight into the pathogenesis of benign and malignant nodules. I have no concerns regarding all of these aspects and believe that this manuscript adds to the current literature. However, I do have the following concerns:

1. In the absence of in vitro or other in-depth follow-up analyses of the GWAS findings, one of the most attractive findings from a clinical perspective is the PRS for benign vs malignant nodules. The authors show that the extreme PRS percentiles are associated with an increased risk of malignant nodules. However, this finding is expected and the real clinical question is whether such a PRS is of additional value to current clinical practice, as it could very well be that it has no real additional value. E.g., how does this PRS compare to the choices made on ultrasound findings (TIRADS scores) and subsequent FNA findings. All of these data should be present in the charts and it would provide much more clarity on the use of these PRS.

TI-RADS has not been used consistently for all thyroid cancer patients in the CCPM Biobank, especially those diagnosed before the year 2017. Equally important, TI-RADS readings are subjective, and accuracy depends on the radiologist's experience. Therefore, in our clinical practice, we always overread the radiologist's interpretation to make a decision to proceed with FNA. To address the Reviewer's question directly and with sufficient scientific rigor, three physicians would need to interpret ~1000 thyroid ultrasounds each and reconcile their readings. This can be done, but the volume of work is very large and can inform a separate publication.

We are confident that PRS adds to thyroid ultrasound interpretation and improves clinical decisions to proceed with the nodule biopsy. We refer the Reviewer to our recent publication (PMID: 37683082), also cited in this revised manuscript, which demonstrated the utility of PRS for thyroid nodule evaluation when PRS was used in combination with AI analysis of thyroid ultrasound images using a convolutional neural network classifier developed by our team. We are currently working on the next generation of this combined classifier, which will incorporate state-of-the-art PRS from this study.

2. The same holds true for the PRS on aggressiveness of thyroid cancer: where would this fit into clinical practice? The authors for example include lymph node metastases, but a neck ultrasound is part of the preoperative workup which is sensitive for detecting lymph node metastases. And if it would be used to detect cancers with higher risk of distant metastases, how does the PRS compare to other conventional tumor characteristics such as pathology characteristics and the presence of lymph node metastases. All of these data should also be present in the charts and it would provide much more clarity on the use of these PRS.

We are grateful to the Reviewer for this suggestion and thoughts. It prompted us to evaluate the role of PRS for detecting aggressive thyroid cancer in a population. Thyroid cancer screening is not recommended by the USPSTF because of the indolent nature of most thyroid cancers and the risk of overdiagnosis. However, we found that PRS can detect patients with high-risk thyroid cancer (as defined by the American Thyroid Association) in a biobank population. The AUC for this use case was superior at 0.741 when compared to other clinical use cases tested.

This discovery may enable a genetically-informed population screening strategy that will identify aggressive thyroid cancers at an early curable stage of development and may significantly reduce morbidity and mortality from thyroid cancer.

The number-needed-to-test (NNT) of 268 for aggressive thyroid cancer screening was comparable to the NNT for the well-established use case of screening colonoscopy for colon cancer (NTT = 262).

We now share clinical data and PRS for all CCPM participants diagnosed with thyroid cancer in the Supplementary Table 15.

In our opinion, this is the most clinically relevant discovery in our study.

In addition to addressing Reviewer's critiques, we made the following changes.

1. We redid all analyses that rely on the updated list of independent loci. This includes mRNA expression analysis, pathway analysis, PRS calculations, etc.

2. We assigned loci as specific to thyroid cancer or benign nodular goiter strictly based on the GWAS significance, removing subjectivity in these group assignments. This led to a shift of some loci from one group to another (e.g., *SDCCAG8* now belongs to loci shared by both thyroid cancer and benign nodular goiter).
3. We assigned pathways to genes based on KEGG annotation and did manual annotation from PubMed search, if the gene is not represented in KEGG. These changes did not affect the primary conclusions of the paper but led to minor updates to Figure 4.
4. We made numerous grammar and style improvements.
5. We updated the introduction and discussion sections to fit the manuscript into the allowed 4000-word limit. Our capacity to add more data/analyses to this study is very limited and will require sacrifices from other sections of the manuscript.

Dear Dr. Danovi and Referees,

Thank you for the favorable review and the acceptance of our manuscript in principle for publication in Nature Genetics.

Please refer to our point-by-point responses to Reviewers' comments, which are highlighted in blue. Technical edits are summarized in the 'NG-A69242R_Pozdeyev_AuthorGuidance_1759979778_1.docx' document.

Nikita Pozdeyev, on behalf of the team of authors.

Reviewers' Comments:

Reviewer #2 (Remarks to the Author):

I thank the authors for addressing my questions satisfactorily including the justification of the applied case-control definition, and for conducting the additional analyses. There are no additional comments or concerns from my side, except that ref 65 in the Methods should also be replaced by 29 (Summix2).

We ensured that the peer-reviewed Summix2 manuscript is cited in the main text and the Methods section.

Reviewer #3 (Remarks to the Author):

I thank the authors for this revised version of the manuscript. Please find below my comments:

Response to my first question: I am satisfied with the authors' response, but I would advise them to include these considerations shortly in the discussion, as I believe that these considerations regarding the place of this PRS in clinical practice will be of interest to the clinical readership.

We cite our study of the PRS in combination with AI in the Discussion (lines 541-544). We added that additional studies are needed to evaluate PRS performance in combination with clinical ultrasound-based risk stratification schemas (lines 544-546).

Response to my second question: I am happy to see that the extra analyses have

shown that this PRS could be of significant use in population screening for aggressive thyroid cancer, which is a real addition to this manuscript.

However, my original question remains on the exact place of this PRS in the workup of a diagnosed thyroid cancer patient: The authors for example include lymph node metastases, but a neck ultrasound is part of the preoperative workup which is sensitive for detecting lymph node metastases. And if it would be used to detect cancers with higher risk of distant metastases, how does the PRS compare to other conventional tumor characteristics such as pathology characteristics and the presence of lymph node metastases? In other words: what is the real addition to clinical practice? In case this remains uncertain, this should be explicitly mentioned in the discussion (while acknowledging the promising findings reported in this manuscript).

We have now discussed two additional potential uses of PRS for clinical decision-making: to improve the performance of molecular tests for managing thyroid nodules with indeterminate cytology and triaging indolent thyroid cancer for active surveillance. We acknowledge that future clinical trials will be needed to determine the exact place of PRS in clinical practice (lines 552-558).

We believe PRS will be inferior to preoperative ultrasound for detecting lymph node metastases and, therefore, did not discuss this application. PRS may also not distinguish clinically meaningful from microscopic lymph node metastases, which may lead to overtreatment.

Similarly, surgical histopathology better predicts the risk of structural disease recurrence. We acknowledge in the revised version of the manuscript that it is not known how PRS can be used in patients already diagnosed with thyroid cancer.

We thank the Reviewer for thoughtful suggestions and for making us think of clinical relevance.